# Traumatic injury compromises nucleocytoplasmic transport and leads to TDP-43 pathology

Eric N Anderson[1], Andrés A Morera[2], Sukhleen Kour[1], Jonathan D Cherry[3,4], Nandini Ramesh[1], Amanda Gleixner[5,6], Jacob C Schwartz[2], Christopher Ebmeier[7], William Old[7], Christopher J Donnelly[5,6], Jeffrey P Cheng[8], Anthony E Kline[8,9], Julia Kofler[10], Thor D Stein[3,4], Udai Bhan Pandey[1,11]*

[1]Department of Pediatrics, Children's Hospital of Pittsburgh, University of Pittsburgh Medical Center, Pittsburgh, United States; [2]Department of Chemistry and Biochemistry, University of Arizona, Tucson, United States; [3]Department of Pathology and Laboratory Medicine, Boston University School of Medicine, Boston, United States; [4]Boston VA Healthcare System, Boston, United States; [5]Department of Neurobiology, University of Pittsburgh School of Medicine, Pittsburgh, United States; [6]LiveLike Lou Center for ALS Research, Brain Institute, University of Pittsburgh School of Medicine, Pittsburgh, United States; [7]Molecular, Cellular & Developmental Biology, University of Colorado, Boulder, United States; [8]Physical Medicine & Rehabilitation; Safar Center for Resuscitation Research, University of Pittsburgh, Pittsburgh, United States; [9]Center for Neuroscience; Center for the Neural Basis of Cognition; Critical Care Medicine, University of Pittsburgh, Pittsburgh, United States; [10]Department of Pathology, University of Pittsburgh, Pittsburgh, United States; [11]Department of Human Genetics, University of Pittsburgh School of Public Health, Pittsburgh, United States

*For correspondence:
udai@pitt.edu

Competing interest: The authors declare that no competing interests exist.

**Abstract** Traumatic brain injury (TBI) is a predisposing factor for many neurodegenerative diseases, including amyotrophic lateral sclerosis (ALS), Alzheimer's disease (AD), Parkinson's disease (PD), and chronic traumatic encephalopathy (CTE). Although defects in nucleocytoplasmic transport (NCT) is reported ALS and other neurodegenerative diseases, whether defects in NCT occur in TBI remains unknown. We performed proteomic analysis on *Drosophila* exposed to repeated TBI and identified resultant alterations in several novel molecular pathways. TBI upregulated nuclear pore complex (NPC) and nucleocytoplasmic transport (NCT) proteins as well as alter nucleoporin stability. Traumatic injury disrupted RanGAP1 and NPC protein distribution in flies and a rat model and led to coaggregation of NPC components and TDP-43. In addition, trauma-mediated NCT defects and lethality are rescued by nuclear export inhibitors. Importantly, genetic upregulation of nucleoporins in vivo and in vitro triggered TDP-43 cytoplasmic mislocalization, aggregation, and altered solubility and reduced motor function and lifespan of animals. We also found NUP62 pathology and elevated NUP62 concentrations in postmortem brain tissues of patients with mild or severe CTE as well as co-localization of NUP62 and TDP-43 in CTE. These findings indicate that TBI leads to NCT defects, which potentially mediate the TDP-43 pathology in CTE.

## Introduction

Traumatic brain injury (TBI) is one of the most common causes of death and disability worldwide (*Zaninotto et al., 2018*). Indeed, secondary injuries resulting from TBI can lead to long-term neurological and neuropsychiatric sequalae, including neurodegenerative diseases such as amyotrophic lateral sclerosis (ALS) (*Chen et al., 2007*; *Peters et al., 2013*), Alzheimer's disease (AD) (*Lye and Shores, 2000*; *Sivanandam and Thakur, 2012*), and Parkinson's disease (*Goldman et al., 2006*; *Harris et al., 2013*; *Taylor et al., 2016*). TBI is also linked with development of chronic traumatic encephalopathy (CTE), a progressive neurodegenerative syndrome associated with repeated head trauma (*Chauhan, 2014*; *Gardner and Yaffe, 2015*; *Stern et al., 2011*). Postmortem brain tissues from repeated trauma patients such as CTE as well as animal models of TBI show microtubule-associated protein (TAU) and TAR DNA/RNA binding protein (TDP-43) pathology (*Blennow et al., 2012*; *Huang et al., 2017*; *Mckee et al., 2018*; *McKee et al., 2010*; *Tan et al., 2018*; *Wang et al., 2015*; *Wiesner et al., 2018*). TDP-43 pathology is a hallmark of neurodegeneration and is present in ~97 % of ALS cases, ~45 % of cases with frontotemporal dementia (FTD) (*Ling et al., 2013*), and ~60 % of AD cases (*Amador-Ortiz et al., 2007*; *Arai et al., 2009*; *Uryu et al., 2008*). Despite evidence of TDP-43 pathology as a biomarker of neurodegeneration, it remains unclear how repeated trauma promotes TDP-43 proteinopathy.

TDP-43 is a predominantly nuclear DNA/RNA-binding protein that shuttles between the nucleus and cytoplasm (*Ayala et al., 2008*). TDP-43 regulates RNA processing such as gene transcription, mRNA splicing, mRNA stability, and mRNA transport and localization, and pathological mutations disrupt RNA metabolism (*Arnold et al., 2013*; *Bose et al., 2008*; *Buratti and Baralle, 2001*; *Polymenidou et al., 2012*). In many neurodegenerative diseases, TDP-43 mislocalizes from the nucleus and aggregate in the cytoplasm of cells. Cytoplasmic TDP-43 aggregates are proposed to be an important mechanistic link to neurodegeneration because these aggregates are abnormally phosphorylated and ubiquitinated (*Arai et al., 2009*; *Arai et al., 2006*; *Neumann et al., 2006*). Multiple mechanisms have been proposed to explain abnormal cytoplasmic accumulation of TDP-43 and progressive spreading of TDP-43 pathology in the context of neurodegenerative diseases.

TDP-43 contains a prion-like low complexity domain and has an intrinsically disordered region (IDR) that renders TDP-43 aggregation-prone (*Calabretta and Richard, 2015*; *Johnson et al., 2009*; *Mann et al., 2019*). IDRs in RNA binding proteins (RBPs) are thought to play key roles in assembly of ribonuclear protein granules such as P-bodies, stress granules, and neuronal granules (*Bakthavachalu et al., 2018*; *Jain et al., 2016*; *Kedersha et al., 2013*), suggesting that mutations or abnormal aggregation of RBPs may disrupt dynamics of these granules (*Molliex et al., 2015*; *Murakami et al., 2015*; *Patel et al., 2015*). Further, IDRs in RBPs such as TDP-43 undergo liquid-liquid phase separation, and pathological mutations in TDP-43 alter liquid-liquid phase separation, which may contribute to disease processes (*Conicella et al., 2016*; *Johnson et al., 2009*; *McGurk et al., 2018*). Alteration in RBPs such as TDP-43 could therefore be an important indicator of neurodegeneration. However, although these mechanisms potentially explain the role of pathological mutations of TDP-43 in the context of neurodegenerative disease, how TDP-43 aggregates form and lead to neurodegeneration in the absence of mutations, such as in the brains of repeated trauma patients, is unclear.

TDP-43 aggregates in ALS/FTD were recently reported to sequester nucleoporins, transport proteins, and other factors, suggesting that TDP-43 aggregation strongly impairs nucleocytoplasmic transport (NCT) and nucleopore complexes (*Chou et al., 2018*). Moreover, NCT is also disrupted in AD (*Eftekharzadeh et al., 2018*), ALS (*Zhang et al., 2015*), and Huntington's disease (HD) (*Gasset-Rosa et al., 2017*; *Grima et al., 2017*), suggesting a common dysfunctional pathway in these neurodegenerative diseases. However, the mechanism of TDP-43 pathology in neurodegeneration resulting from repeated head trauma is unknown. We previously demonstrated that repetitive trauma leads to ubiquitin, p62, and TDP-43 inclusion as well as stress granule pathology in *Drosophila* brains (*Anderson et al., 2018*). Here, we performed proteomic analysis on *Drosophila* brains to identify the molecular pathways that were altered in response to traumatic injury.

In this manuscript, we found that repeated trauma upregulates nuclear pore proteins, alter nucleoporin stability, and NCT proteins as well as alters RanGAP1 and nucleoporins distribution, and NCT in vivo. In addition, pharmacological inhibition of nuclear export protects against TBI-mediated lethality and NCT defects. Intriguingly, upregulation of nucleoporins in vivo and in vitro leads to TDP-43 mislocalization, aggregation, phosphorylation, and alteration in solubility, and decreased motor function as well as lifespan. Our findings of NUP62 pathology and increased NUP62 concentrations in brain

tissues from patients with CTE suggest that NCT defects are linked with traumatic injury, which potentially mediates TDP-43 pathology.

## Results

### TBI alters the *Drosophila* brain proteome, perturbs NCT proteins, and Nup protein level

To examine the mechanism of TBI, we used a well-characterized *Drosophila* model of TBI that exhibits robust phenotypes such as TDP-43 homolog (Tbph), p62, stress granules, and ubiquitin pathology (*Anderson et al., 2018*; *Katzenberger et al., 2015*; *Katzenberger et al., 2013*) to identify and investigate changes in the brain proteome post-TBI (*Figure 1A*). An unbiased proteomics analysis of 2000 proteins from third instar control (*w1118*) *Drosophila* larval brains exposed to repeated TBI identified 361 proteins that were significantly changed in response to traumatic injury (p ≤ 0.05, student's T-test). Complex network analysis based on gene ontology (GO) association analysis in BiNGO (Cytoscape 3) of the TBI-associated brain proteome compared to non-TBI controls identified distinct categories of altered proteins, with the majority upregulated (*Figure 1B and C*; *Figure 1—figure supplement 1A-E*; *Supplementary files 1 and 3*). Top categories that were upregulated after TBI were microtubule cytoskeleton, protein folding, and the proteasome. In addition, we also identified proteins involved in the ribonuclear protein complex and spliceosome. Components of the nuclear pore complex (NPC) and NCT pathway were a major subset of proteins upregulated post-TBI (*Figure 1—figure supplement 1F*). The NPC pathway has not previously been linked to trauma-mediated neurodegeneration but has been reported to be disrupted in ALS/FTD, AD, HD, and other neurodegenerative diseases (*Chou et al., 2018*; *Eftekharzadeh et al., 2018*; *Grima et al., 2017*; *Zhang et al., 2015*). Therefore, we decided to further examine the NPC alteration in mediating TDP-43 pathology in TBI.

NPC is a large multi-protein complex that spans the nuclear envelope and mediates NCT. In vertebrates, NPC comprises multiple copies of ~30 proteins called nucleoporins (Nups) with highly conserved function between species (*Nofrini et al., 2016*). Our proteomics analysis revealed that several Nups were upregulated in response to TBI in *Drosophila* brains (*Figure 1D and E*). Quantitative reverse-transcriptase polymerase chain reaction (qRT-PCR) analysis of these Nups confirmed that TBI significantly upregulated *Nup93-2* (p < 0.01), *Nup54* (p < 0.001), *Nup62* (p < 0.01), *Nup44A* (p < 0.05), and *Nup214* (p < 0.05) mRNA levels in *Drosophila* larval (*Figure 1F–I and − Figure 1 − Figure Supplement 1G*) and adult brains (*Figure 1J–L*; *Nup62,* p < 0.01; *Nup214*, p < 0.01; *Nup54, p < 0.001*). In addition, mRNA levels of nuclear export gene *Emb* (*Exportin*) were significantly increased post-TBI (*Figure 1M*, p < 0.001), while the microtubule-associated protein *Futsch* showed no change in mRNA levels compared to controls (*Figure 1 − Figure Supplement 1H*), consistent with proteomics analysis. Western blotting further confirmed a TBI-dependent increase in *Drosophila* Nup214 protein levels in larval brains (*Figure 1N and O*).

We next asked if trauma alter endogenous nucleoporin protein levels overtime. To address this, we measured Nup214 protein levels in trauma and control larval and adult brains. We performed time-course analysis on larval (0, 2, 4, and 6 hr) or adult (0, 2, 4, 24, and 72 hr) animals' post-injury to assess Nup214 protein level by Western blot analysis. Interestingly, we found that Nup214 protein level remains upregulated in both larval and adult brains over the time point examined (*Figure 1P,Q,R,S*), suggesting that trauma probably disrupts Nup214 levels and possibly turnover in vivo.

### TBI disrupts NPC and RanGAP1 distribution and leads to TBPH/NPC coaggregation

To test whether repeated trauma disrupts NPC, we examined the distribution on the nuclear membrane of TBI *Drosophila* brains. Using the NPC marker Mab414, which recognizes the FG domains of several Nups including Nup62, we found a predominantly homogenous rim-like labeling of the nuclear membrane in non-TBI control *Drosophila* brains. NPC staining in the ventral nerve cord (VNC) in non-TBI control brains appeared distributed evenly. In contrast, nuclear membrane NPC staining in VNCs exposed to TBI was irregular and revealed a disturbed nuclear morphology. We observed gaps in the nuclear membrane and the appearance of aggregation (Mab414 clumps) in brains post-TBI (*Figure 2A*). Quantitatively, the percentage of cells with abnormal Mab414 staining was significantly higher in TBI brains than non-TBI controls (*Figure 2B*, p < 0.001), suggesting that traumatic injury

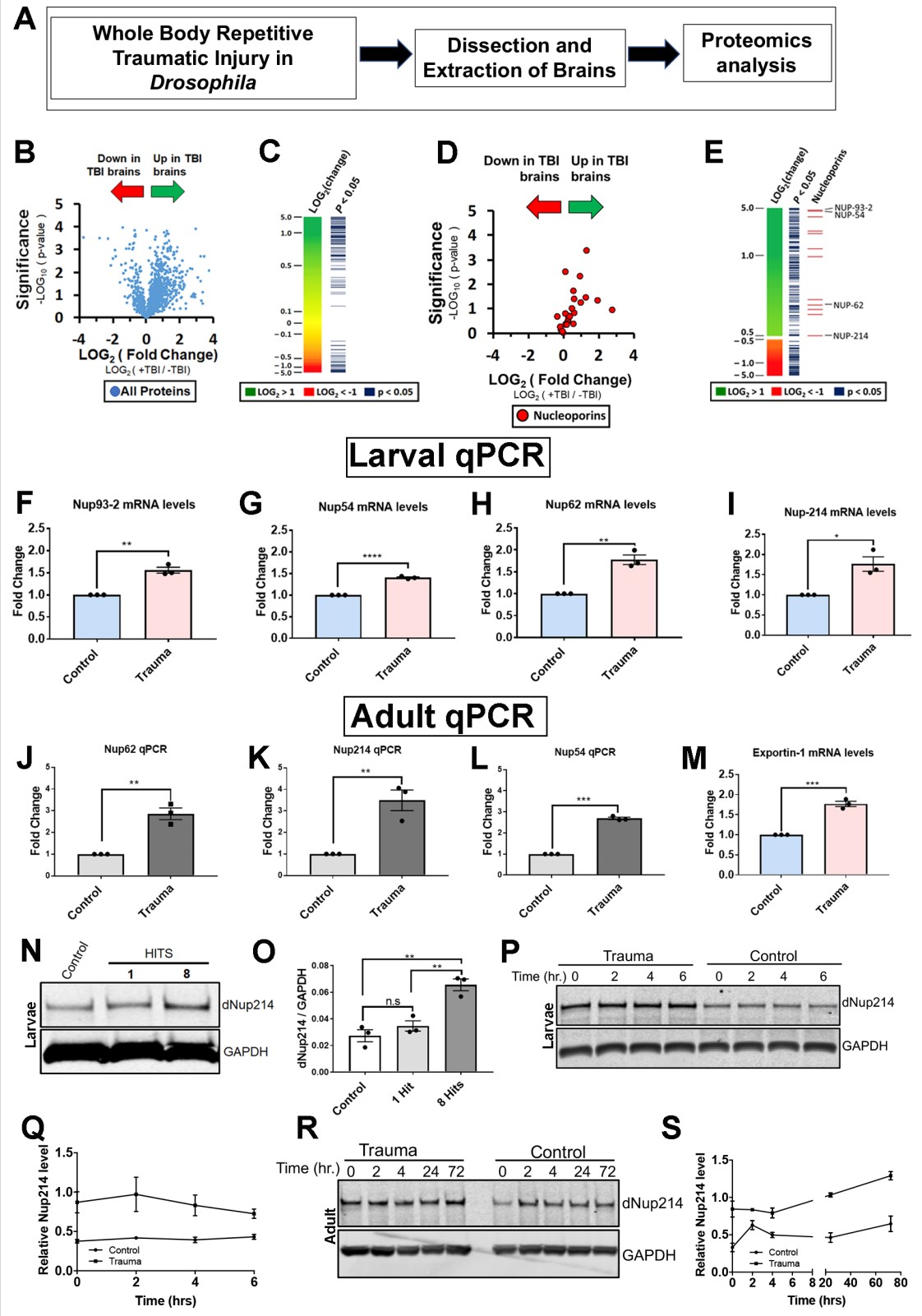

**Figure 1.** Proteomic analysis of *Drosophila* brains identifies novel components of the nucleocytoplasmic transport machinery that are disrupted in response to TBI. (**A**) Schematic flow of *Drosophila* larval brain traumatic injury, dissection, and proteomics analysis. (**B**) Volcano plot showing fold-change and *p*-values for all detected proteins (student's t-test, assuming equal variances). (**C**) Heat map showing change in protein expression for 2000 detected proteins. Proteins with a significant change (p < 0.05) are indicated. (**D**) Volcano plot showing fold change and *p*-values of all nucleoporins and nuclear

*Figure 1 continued on next page*

*Figure 1 continued*

transport proteins. Traumatic injury upregulates components of the nuclear pore and nucleocytoplasmic transport. (**E**) Heat map and p-values for proteins with more than ±1.4-fold change with the indicated nucleoporins. (**F–I**) Quantitative reverse-transcriptase polymerase chain reaction (qRT-PCR) analysis of nucleoporins identified in the proteomics analysis showed significant increases in (**F**) *Nup93-2*, (**G**) *Nup54*, (**H**) *Nup62*, and (**I**) *Nup214* mRNA levels in *w^1118* *Drosophila* larval brains exposed to trauma (eight hits @ 50° angle) compared to control animals (n = 3, *p < 0.05, **p < 0.01, ****p < 0.0001). (**J–L**) qPCR analysis of *Nup62* (**J**, **p < 0.01), *Nup214* (**K**, **p < 0.01), and *Nup54* (**L**, **p < 0.001) mRNA levels in adult *Drosophila* brains showed a significant increase in response to trauma as compared to control brains (n = 3). (**M**) qRT-PCR analysis of nuclear export factor *Emb* (exportin) mRNA post-trauma compared to non-trauma controls (n = 3, ***p < 0.001). (**N**) Western blot of *Drosophila* larval brains exposed to trauma (one or eight hits @ 50° angle) and controls (0 hits) probed for Nup214 protein. *Drosophila* Gapdh is shown as a loading control. (**O**) Quantification of western blots of *Drosophila* Nup214 protein levels compared to controls (n = 3, **p < 0.01, ***p < 0.001, n.s. = not significant). (**P**) Western blot of Nup214 protein level in larval brains at 0, 2, 4, and 6 hr post-injury or controls. (**Q**) Quantification of Nup214 protein levels in larval brains of trauma and control at 0, 2, 4, and 6 hr (n = 3). (**R**) Western blot of Nup214 protein level in adult brains at 0, 2, 4, 24, and 72 hr post-injury or controls. (**S**) Quantification of Nup214 level levels in adult brains of trauma and control at 0, 2, 4, 24, and 72 hr (n = 3). All qRT-PCR and western blot analysis were done in triplicate using biological replicates. One-tailed t-test was used in panels F–L and O. One-way ANOVA with Tukey's multiple comparisons tested was used for panel N. All quantification represents mean ± s.e.m.

The online version of this article includes the following figure supplement(s) for figure 1:

**Figure supplement 1.** Proteomic analysis of other altered proteins in response to traumatic injury.

disrupts NPC morphology. Next, we asked whether Tbph aggregation identified in our *Drosophila* larval and adult model of trauma (*Anderson et al., 2018*) co-aggregate with NPC (Mab414). We found perinuclear and cytoplasmic coaggregation of Tbph and Mab414 in TBI-exposed VNCs, while control brains showed little to no coaggregation (*Figure 2C*). Quantification showed that the percentage of Mab414–Tbph-positive cells with coaggregation was significantly higher in TBI-exposed VNCs compared to controls (*Figure 2D*, p < 0.001).

Ran GTPase-activating protein 1 (RanGAP1), which is located on cytoplasmic filaments of the NPC, is required for Ran-dependent nuclear import and export (*Bischoff et al., 1995*). GTP hydrolysis of Ran-GTP by RanGAP1 is essential for cargo release into the cytoplasm during nuclear export (*Floch et al., 2014*). RanGAP1 maintains the nuclear/cytoplasmic Ran gradient, and its loss causes cell death (*Hetzer et al., 2002*). Intriguingly, RanGAP1 gradient is disrupted in neurodegenerative diseases including ALS (*Zhang et al., 2015*) and HD (*Grima et al., 2017*). Therefore, we examined whether repeated trauma influences RanGAP1 distribution in the VNCs of *Drosophila* larvae. RanGAP1 in non-TBI control brains appeared mostly uniform in distribution within the nuclear membrane, with few cells showing intense nuclear signals. However, TBI-exposed brains had aberrant distribution of RanGAP1 staining, with intense nuclear and cytoplasmic intensity (*Figure 2E*, arrows). The percentage of cells with abnormal RanGAP1 distribution was significantly higher in brains post-TBI compared to controls (*Figure 2F*, p < 0.001), suggesting that repeated traumatic injury disturbs RanGAP1 localization in the brain. TBI-mediated disruption in NPCs or RanGAP1 was not due to apoptosis, as TUNEL assays showed no elevation in cell death immediately after injury (*Figure 2—figure supplement 1A,B*).

To further determine whether RanGAP1 and Nup mislocalization and/or aggregation occurs in mammals, we assessed distribution in a control cortical impact (CCI) rat model of TBI (*Xiong et al., 2013*; *Figure 2G*). Brain cells in hippocampal regions underlying the injured cortex (ipsilateral hemisphere) of rats with TBI exhibited either mislocalization and/or aggregation (arrows) or intense nuclear staining (arrow heads) of RanGAP1, compared to predominantly smooth perinuclear staining in sham controls (*Figure 2H*). Quantitatively, the percentage of cells with mislocalization and/or aggregation of RanGAP1 was significantly higher in TBI brains than controls (*Figure 2I*, p < 0.01). In contrast, RanGAP1 pathology in the contralateral hemisphere of trauma brains showed a mild increase that was not significant compared to controls (*Figure 2—figure supplement 2A,B*).

We also evaluated whether nucleoporins exhibited mislocalization and/or aggregation by staining for NUP62. Although we did not detect many instances of NUP62 aggregation, NUP62 showed dramatic cytoplasmic (arrows) and nuclear (arrow heads) mislocalization in TBI rats compared to sham controls, which showed few instances of cytoplasmic but no nuclear mislocalization (*Figure 2J*). Quantitatively, the percentage of cells with NUP62 pathology was significantly higher in TBI brains compared to sham controls (*Figure 2K*, p < 0.01). Similarly, the percentage of cells with NUP62 pathology contralateral to the injury site was significantly higher in TBI brains compared to controls (*Figure 2—figure supplement 2C,D*, p < 0.05).

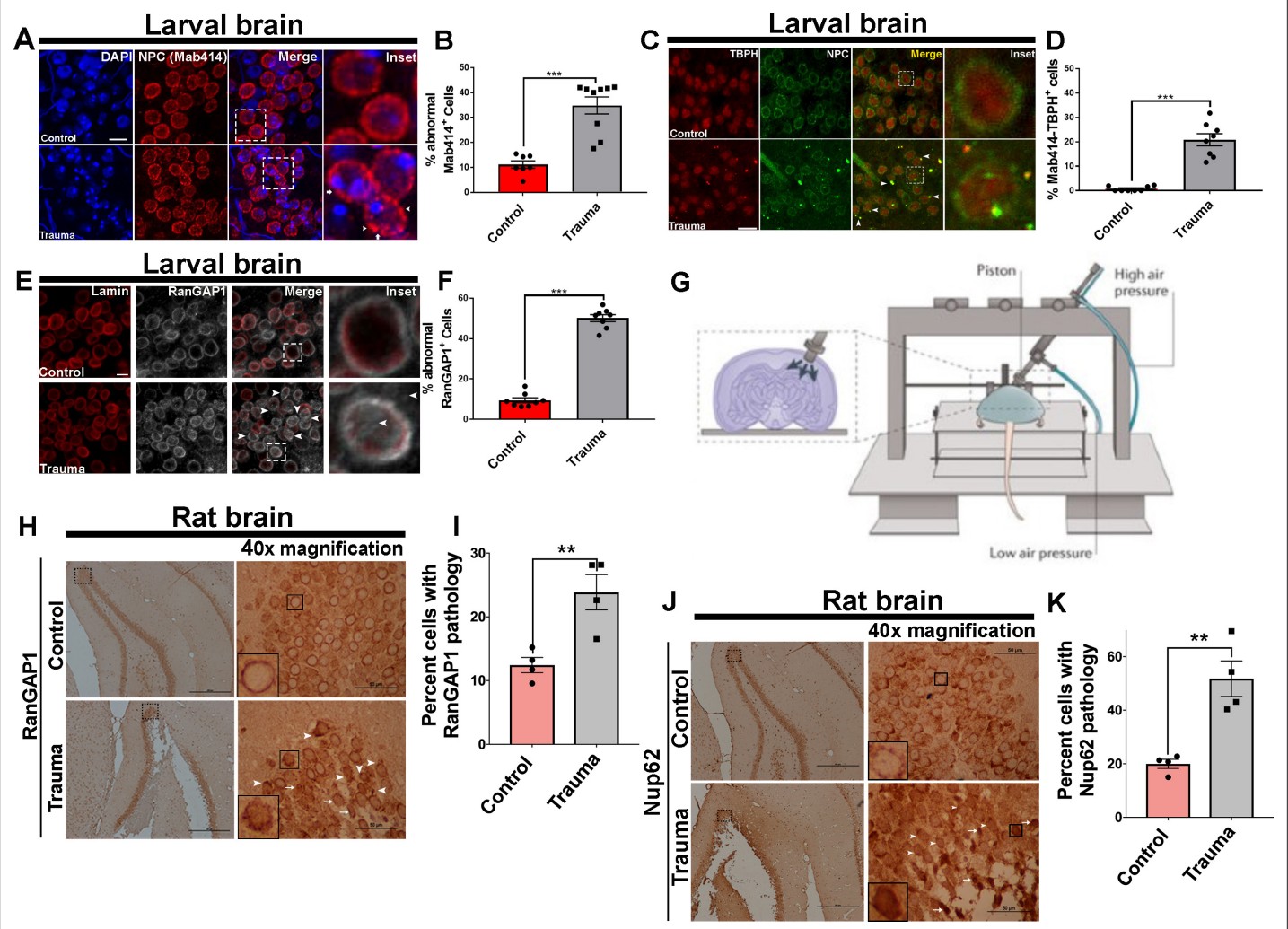

**Figure 2.** Nuclear pore complex morphology and RanGAP1 distribution defects as well as TBPH/NPC coaggregation after traumatic injury in vivo. (**A**) Immunofluorescence images of the ventral nerve cord (VNC) of trauma brain injury (Trauma) and non-TBI control $w^{1118}$ *Drosophila* larvae stained with the nuclear pore complex (NPC) marker (Mab414, red) marker and nuclei (DAPI, blue). TBI NPCs show gaps in membranes (arrows) with signs of aggregation (arrow heads). Scale bar = 15 µM. (**B**) Quantification of percentage of VNC cells with abnormal (Gaps and aggregation) Mab414 staining after TBI compared to non-TBI controls (n = 7–9 larval VNCs, ***p < 0.001). (**C**) Representative images of larval VNCs in TBI and control brains stained for the *Drosophila* homolog of TDP-43, TAR DNA-binding protein-43 homolog (Tbph, red), and Mab414 (green), showing perinuclear as well as cytoplasmic coaggregation (arrows, yellow). Scale bar = 20 µM. (**D**) Percentage of cells with Tbph/Mab414 coaggregation in the VNCs of TBI and control brains (n = 8 larval VNCs, ***p< 0.001). (**E**) Immunofluorescence of RanGAP1 (gray) and Lamin (red) in TBI and control VNCs shows aberrant RanGAP1 staining in TBI brains (intense nuclear and cytoplasmic staining, arrows). Scale bar = 15 µM. (**F**) Percentage of cells with abnormal RanGAP1 staining in TBI VNCs compared to controls (n = 6 larval VNCs, ***p < 0.001). (**G**) Schematic of control cortical impact (CCI) model of TBI (adopted from *Xiong et al., 2013*). (**H**) Immunohistochemical staining of RanGAP1 in rat hippocampal region (granule cell layer and subgranular zone) underlying the cortex in TBI animals or controls shows intense nuclear staining (arrow heads) and cytoplasmic aggregation (arrows). Inset box shows ×40 magnification (scale bar = 50 µM). (**I**) Percentage of cells with RanGAP1 pathology (cytoplasmic/nuclear aggregate or intense nuclear/cytoplasmic mislocalization) in TBI brains and controls (n = 4, **p < 0. 01). (**J**) Immunohistochemical staining of Nup62 in rat hippocampal regions shows intense nuclear staining (arrow heads) and cytoplasmic aggregation (arrows). Inset box shows ×40 magnification (scale bar = 50 µM). (**K**) Percentage of cells with Nup62 pathology (cytoplasmic/nuclear aggregate or intense nuclear/cytoplasmic mislocalization) in TBI brains and controls (n = 4, **p < 0.01). One-tailed t-tested was used in panel B, D, F, I, and K. Data represent mean ± s.e.m.

The online version of this article includes the following figure supplement(s) for figure 2:

**Figure supplement 1.** Repeated traumatic injury does not lead to cell death immediately following injury.

**Figure supplement 2.** RanGAP1 and NUP62 distribution is mildly altered in the contralateral hemisphere in a CCI trauma rat model.

## TBI-mediated NCT defects and lethality are suppressed by nuclear export inhibitors in vivo

To determine whether TBI impairs nucleocytoplasmic transport in vivo, we overexpressed a GFP protein tagged with a nuclear localization sequence (NLS) and nuclear export sequence (NES) (NLS-NES-GFP) (*Kusano et al., 2001*) in *Drosophila* motor neurons (OK371-gal4). GFP signals in NLS-NES-GFP-expressing brains appeared distributed in both the nucleus and cytoplasm (*Figure 3A*). However, quantitative analysis showed that the percentage of cells with reduced nuclear GFP signals in NLS-NES-GFP-expressing VNCs exposed to TBI was significantly increased while nuclear GFP intensity was significantly reduced (*Figure 3B and C*, p < 0.001), suggesting that nuclear import of GFP was inhibited and/or nuclear export of GFP was increased. We next expressed a GFP protein tagged with an NLS and mutated NES (ΔNES) without nuclear export activity (*Kusano et al., 2001*) in motor neurons. Brain cells of non-TBI animals expressing NLS-ΔNES-GFP showed robust GFP localization to the nucleus (*Figure 3A*), whereas TBI animal VNCs showed significantly increased number of cells with reduced nuclear GFP (p < 0.01) and decreased nuclear GFP intensity (p < 0.001) (*Figure 3B and C*). Together, these data strongly suggest that traumatic injury impair nuclear import.

Previously, we showed that repeated trauma in *Drosophila* larvae leads to lethality (*Anderson et al., 2018*). To assess whether pharmacological inhibition of trauma-mediated NCT defects protects against lethality, we fed selective inhibitors of nuclear export (SINE) compounds KPT-350 or KPT-276 to *Drosophila* larvae post-injury. KPT-350 and KPT-276 show neuroprotection in C9orf72 mediated ALS neurons and *Drosophila* models *Tamir et al., 2017*; *Zhang et al., 2015*; KPT-350 reduce cell death in rodent primary cortical neurons expressing HD protein *Grima et al., 2017*; and KPT-350 protect cortical circuit function and survival of inhibitory interneurons as well as reduce TBI-induced behavioral and histological deficits in a CCI model of TBI (*Cantu et al., 2016*; *Tajiri et al., 2016*). *Drosophila* larvae were exposed to TBI and raised on either DMSO alone, KPT-350 (0.05, 0.1, or 0.5 mM), or KPT-276 (50, 200, or 1000 nM). Eclosion assays on *Drosophila* larvae that experienced TBI showed a significant dose-dependent suppression of lethality in animals treated with KPT-350 (*Figure 3D*, **p < 0.01) or KPT-276 (*Figure 3E*, *p < 0.05, ***p < 0.001). Because KPT-350 and KPT-276 rescue aberrant NCT in *Drosophila* models of C9 ALS/FTD (*Tamir et al., 2017*), we examined whether administration of these compounds also protected *Drosophila* against TBI-mediated NCT defects. *Drosophila* larvae expressing NLS-NES-GFP or NLS-ΔNES-GFP in motor neurons, exposed to TBI, and fed KPT-350 (0.5 mM) for 24 hr showed strong nuclear GFP intensity compared to DMSO-treated controls (*Figure 3F*, ***p < 0.001). Quantitatively, nuclear GFP signals in KPT-350-treated flies were significantly higher than DMSO-treated controls (*Figure 3G*, p < 0.001). KPT-350 treatment in adult *Drosophila* exposed to TBI showed a dose-dependent suppression of Tbph puncta post-trauma (*Figure 3—figure supplement 1A,B*, p < 0.05). Taken together, these results suggest that suppression of trauma mediated NCT defects and/or Tbph cytoplasmic puncta via inhibition of nuclear export correlate with increased survival.

## NUPs upregulation causes TDP-43/TBPH mislocalization and aggregation in vivo and in vitro

Upregulation of Nups in our proteomics analysis and TDP-43 aggregation post-TBI (*Anderson et al., 2018*) prompted us to investigate whether Nup upregulation can directly lead to TDP-43 mislocalization. Previous studies report TDP-43 pathology in postmortem brain tissues of patients who experience repeated trauma (*Mckee et al., 2018*; *McKee et al., 2010*) as well as in animal models of TBI (*Huang et al., 2017*; *Tan et al., 2018*; *Wiesner et al., 2018*). Consistent with this, we recently showed that repeated traumatic injury in larval and adult *Drosophila* leads to stress granule induction, ubiquitin, p62 (Ref2(P)), and Tbph pathology (*Anderson et al., 2018*). Moreover, TDP-43 droplets were recently shown to recruit *NUP62* and induce mislocalization of RanGAP1 and Nup107 (*Gasset-Rosa et al., 2019*), suggesting that NUPs might be important modulators of TDP-43 proteinopathy. To test the impact of Nup upregulation on Tbph (*Drosophila* TDP-43) pathology, we generated a site-specific *Nup62* overexpression (Nup62 OE) fly line and stained for endogenous Tbph. The distribution of Tbph in eGFP control (henceforth referred to as control) expressing *Drosophila* larval VNC appeared predominantly nuclear, while Nup62 OE larval VNC had dramatically altered Tbph localization with cytoplasmic aggregation (*Figure 4A*). Quantitative analysis revealed a significant increase in Tbph puncta in Nup62 OE *Drosophila* larvae or adult brains compared to controls (*Figure 4B*, p < 0.001;

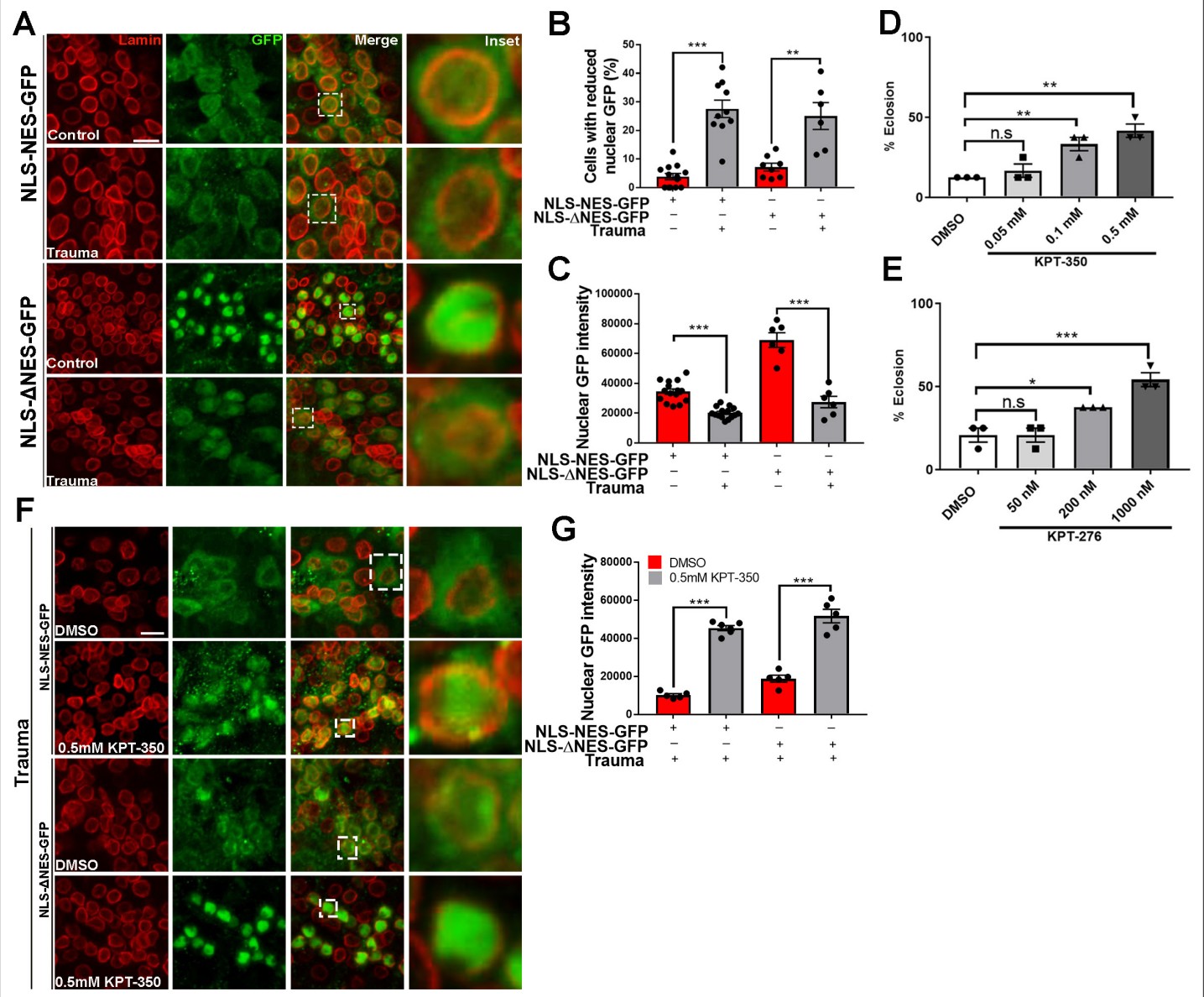

**Figure 3.** Pharmacological suppression of nuclear export rescues trauma-mediated nucleocytoplasmic transport defects and lethality in vivo. (**A**) Representative immunofluorescence images of larval ventral nerve cord (VNC) cells expressing NLS-NES-GFP or NLS-ΔNES-GFP in motor neurons (OK371-gal4) exposed to traumatic brain injury (TBI) or controls (non-TBI) were co-stained for anti-GFP (green) and nuclear envelope marker Lamin (red). Scale bar = 15 μM. (**B**) Percentage of cells with reduced nuclear GFP signal in animals expressing NLS-NES-GFP (n = 10–11) or NLS-ΔNES-GFP GFP (n = 6–10) exposed to TBI or non-TBI controls (**p < 0.01, ***p < 0.001). (**C**) Quantification of nuclear GFP signal intensity in TBI and non-TBI NLS-NES-GFP (n = 15) or NLS-ΔNES-GFP (n = 6) expressing animals (***p < 0.001). (**D, E**) Quantification of eclosion assay of trauma *Drosophila* larvae treated with DMSO only compared to those treated with (**D**) KPT-350 (0.05, 0.1, or 0.5 mM) or (**E**) KPT-276 (50, 200, or 1000 nM). Treatment with KPT-350 or KPT-276 shows a dose-dependent rescue of trauma-mediated lethality (n = 3; *p < 0.05, **p < 0.01). Results indicate one-way ANOVA with Tukey's multiple comparisons. All quantifications represent mean ± s.e.m. (n = 15 animals per experiment). (**F**) Representative immunofluorescence confocal images of *Drosophila* VNCs expressing NLS-NES-GFP or NLS-ΔNES-GFP in motor neurons exposed to TBI and fed KPT-350 (0.5 mM) or DMSO alone for 24 hr show that KPT-350 protects against trauma-mediated depletion of nuclear GFP signal. Scale bar = 15 μM. (**G**) Quantification of nuclear GFP signal intensity in animals expressing NLS-NES-GFP or NLS-ΔNES-GFP exposed to TBI or non-TBI controls (n = 5–6, ***p < 0.001).

The online version of this article includes the following figure supplement(s) for figure 3:

**Figure supplement 1.** Pharmacological suppression of nuclear export rescues trauma mediated Tbph aggregation.

*Figure 4—figure supplement 1A,B*, p < 0.001). The Tbph puncta in Nup62 OE VNCs congregated with NPC/Mab414 (*Figure 4—figure supplement 2A*). We further showed that nuclear Tbph intensity

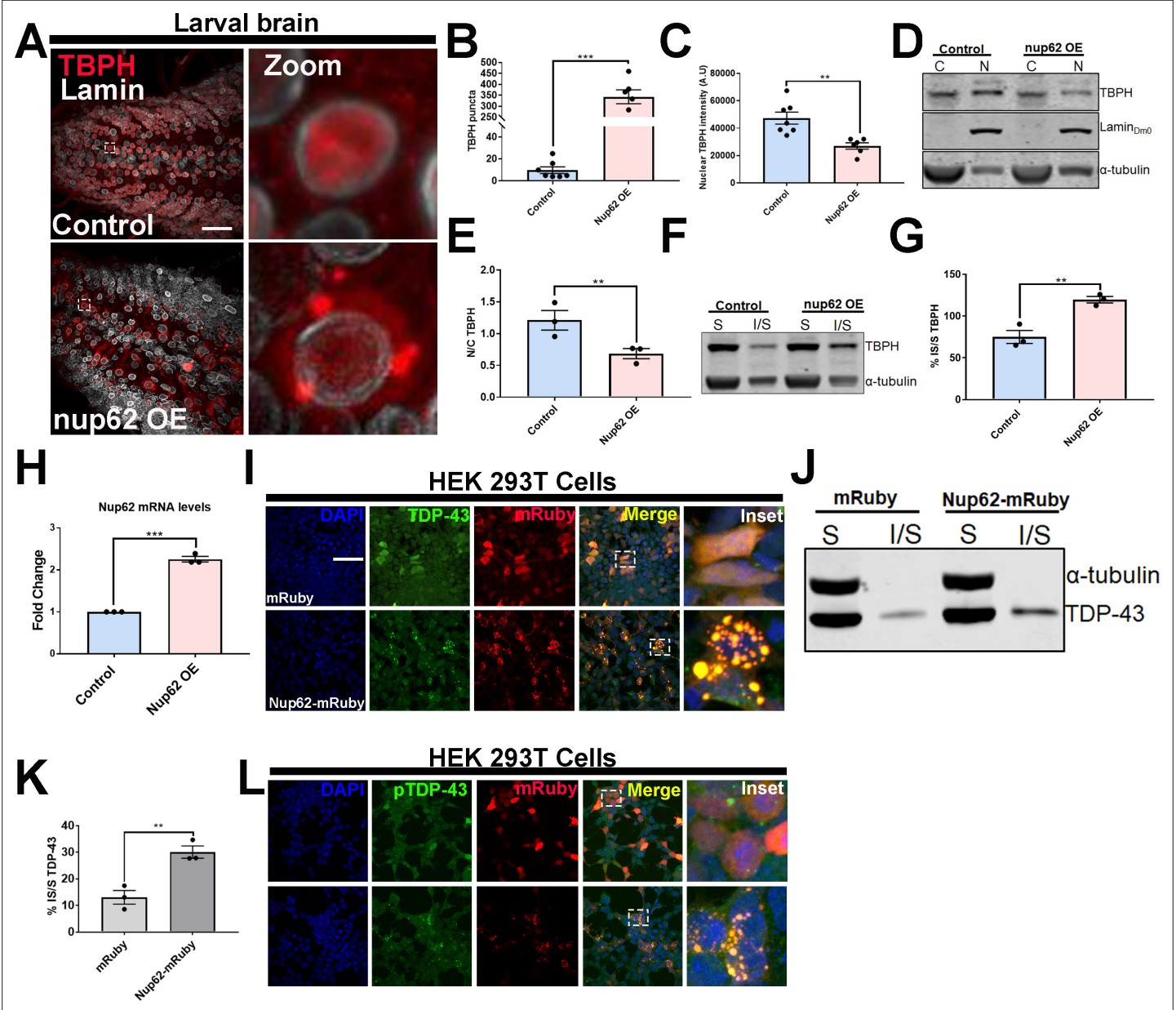

**Figure 4.** Nup62 expression promotes TDP-43 mislocalization and aggregation in vivo and in vitro. (**A**) *Drosophila* larval brain overexpressing Nup62 (*Nup62* OE) in motor neurons (Ok371-gal4) and controls (eGFP) stained for the *Drosophila* homolog of TDP-43, TAR DNA-binding protein-43 homolog (Tbph), and the nuclear envelope marker Lamin, showing Tbph accumulation. Scale bar = 25 μM. Zoom image represents inset box. (**B**) Quantification of the number of Tbph puncta in Nup62 OE flies compared to controls (n = 5–7, ***p< 0.001). (**C**) Nuclear Tbph intensity in Nup62 OE flies compared to controls (n = 5–7 brains, **p < 0.01). (**D**) Western blots of cytoplasmic (**C**) and nuclear (**N**) fractions from *Drosophila* expressing *Nup62* OE in motor neurons and control probed for Tbph, lamin, and tubulin. (**E**) Nuclear-cytoplasmic (N/C) ratio quantification of Tbph in Nup62 OE and control animals (n = 3 blots, **p < 0.01). (**F**) Western blots of soluble (**S**) and insoluble (I/S) fractions *Drosophila* expressing *Nup62* OE in motor neurons and controls probed for Tbph and tubulin. (**G**) Percentage of Tbph solubility in Nup62 OE and control motor neurons (n = 3 blots, **p < 0.01). (**H**) qRT-PCR analysis of *Nup62* mRNA in Nup62 OE animals compared to controls (n = 3, ***p < 0.001). (**I**) Representative immunofluorescence images of human embryonic kidney 293T (HEK293T) cells transfected with mRuby alone (red) and NUP62-mRuby show endogenous TDP-43 (green) coaggregation with NUP62. NUP62-mRuby co-localizes with endogenous TDP-43 (merge, yellow). Scale bar = 25 μM. (**J**) Western blots of soluble and insoluble fractions from HEK293T cells transfected with mRuby alone or NUP62-mRuby probed for TDP-43 and tubulin. (**K**) Quantification of TDP-43 solubility in HEK293T cells transfected with mRuby alone or NUP62-mRuby (n = 3 blots, **p < 0.01). (**L**) Representative immunofluorescence images of HEK293T cells transfected with NUP62 show phosphorylated TDP-43 accumulation (green) that co-localizes with NUP62 (merged, yellow). Scale bar = 25 μM. qRT-PCR and western blot analysis were done in triplicate using biological replicates. One-tailed t-test was used in panel B, C, E, G, H, and K. All quantification represent mean ± s.e.m.

*Figure 4 continued on next page*

*Figure 4 continued*

The online version of this article includes the following figure supplement(s) for figure 4:

**Figure supplement 1.** Nup62 expression modulates TBPH and TDP-43 aggregation as well as mislocalization.

**Figure supplement 2.** Expression of Nup62 but not Nup214 or Nup43 leads to NPC/TBPH coaggregation.

**Figure supplement 3.** Increased Nup93-2 but not Nup44A expression partially leads to TBPH mislocalization and aggregation in vivo.

**Figure supplement 4.** *Nup*214 and *Nup*43 expression leads TDP-43 mislocalization and aggregation.

**Figure supplement 5.** NUP54 expression does not alter TDP-43 localization.

**Figure supplement 6.** Expression of other upregulated proteins in the top five categories does not alter TDP-43 localization.

in Nup62 OE VNCs was significantly reduced compared to controls (*Figure 4C*, p < 0.01). Moreover, nuclear-cytoplasmic (N/C) fractionation further confirmed a significantly reduced Tbph N/C ratio in Nup62 OE animals compared to controls (*Figure 4D and E*, p < 0.01), suggesting that *Nup62* upregulation alters the subcellular localization of Tbph in vivo. Mislocalization and aggregation of Tbph caused by Nup62 expression also led us to examine if Tbph solubility was affected by Nup62 expression. Soluble-insoluble fractionation of Nup62 OE or control *Drosophila* brains showed that upregulation of Nup62 expression in motor neurons shifted Tbph solubility by significantly increasing insolubility compared to controls (*Figure 4F and G*, p < 0.01), suggesting that Nup62 levels are an important determinant of Tbph solubility. Overexpression of *Nup62* was confirmed by qPCR (*Figure 4H*, p < 0.001).

We further evaluated the impact of Nup62 overexpression on human TDP-43 protein in *Drosophila*. We crossed Nup62 OE flies to a previously published line made using CRISPR/Cas9 genome editing that removed the endogenous Tbph gene and replaced it with FLAG-tagged wild-type human TDP43 (hTDP-43WT) (*Chang and Morton, 2017*). FLAG staining in CRISPR hTDP-43WT control *Drosophila* brains appeared nuclear, with few puncta. However, Nup62 OE; CRISPR hTDP-43WT brains had significantly increased FLAG puncta (p < 0.01) and reduced nuclear TDP-43 intensity (p < 0.05) compared to controls (*Figure 4—figure supplement 1C,D,E*), further suggesting that Nup62 overexpression disrupts TDP-43 subcellular localization.

We also expressed four additional Nups that were upregulated in our proteomic analysis, EP (Enhancer-promoter) overexpression Nup93-2 and Nup44A lines, Nup214 OE, and HA-tagged Nup43 OE, to determine the contribution of other Nups on Tbph subcellular distribution. There was a significant increase in Tbph puncta in flies overexpressing Nup93-2 (p < 0.001) but not Nup44A (*Figure 4—figure supplement 3A,B*). Further, Nup93-2 and Nup44A overexpressing flies showed a trend of decreasing nuclear Tbph intensity that was not significant (*Figure 4—figure supplement 3C*). qPCR analysis validated that flies significantly overexpressed *Nup93-2* and *Nup44A* mRNA (*Figure 4—figure supplement 3D,E*, p < 0.001). Nup214 or Nup43 overexpression led to Tbph aggregation (*Figure 4—figure supplement 4A,B*, p < 0.001) and decreased N/C Tbph ratio (*Figure 4—figure supplement 4C,D*, p < 0.05), although the Tbph aggregates in Nup214 and Nup43-expressing animals did not localize with Mab414 (*Nup214*) or Anti-HA (*Nup43*) (*Figure 4—figure supplement 2A,B*). qPCR analysis showed significant increase *Nup214* and *Nup43* mRNA levels (*Figure 4—figure supplement 4E,F*, p < 0.001). Together, these data suggest that alteration in multiple Nups may contribute to Tbph mislocalization and/or aggregation, albeit with a more pronounced effect by Nup62.

We next assessed the effects of NUP62 expression on endogenous TDP-43 localization and aggregation in human embryonic kidney cells (HEK293T). HEK293T cells were transfected with mRuby alone or mRuby tagged NUP62 (Nup62-mRuby). Expression of NUP62 in HEK293T led to aggregation of the mRuby reporter (*Figure 4I*). FG-Nups such as NUP62, NUP214, and NUP54 are predicted to contain prion-like domain and a low complexity domain, which is suggested to mediate cytoplasmic coaggregation with the C-terminal fragment of TDP-43 (TDP-CTF) (*Chou et al., 2018*). Interestingly, NUP62 co-aggregated with endogenous TDP-43 (*Figure 4I*). Moreover, expression of NUP62 in HEK293T cells led to a significant shift in TDP-43 solubility compared to mRuby alone (*Figure 4J and K*, p < 0.01), similar to the *Drosophila* Nup62 overexpression model. TDP-43 aggregates are phosphorylated and are toxic in vitro and in vivo (*Arai et al., 2009*; *Choksi et al., 2014*; *Zhang et al., 2009*),. Therefore, we assessed whether NUP62-mediated TDP-43 aggregates were phosphorylated in HEK293T cells. Endogenous TDP-43 aggregates that co-localized with NUP62-mRuby were phosphorylated (*Figure 4L*), suggesting that TDP-43 aggregates might be pathological and toxic. Interestingly,

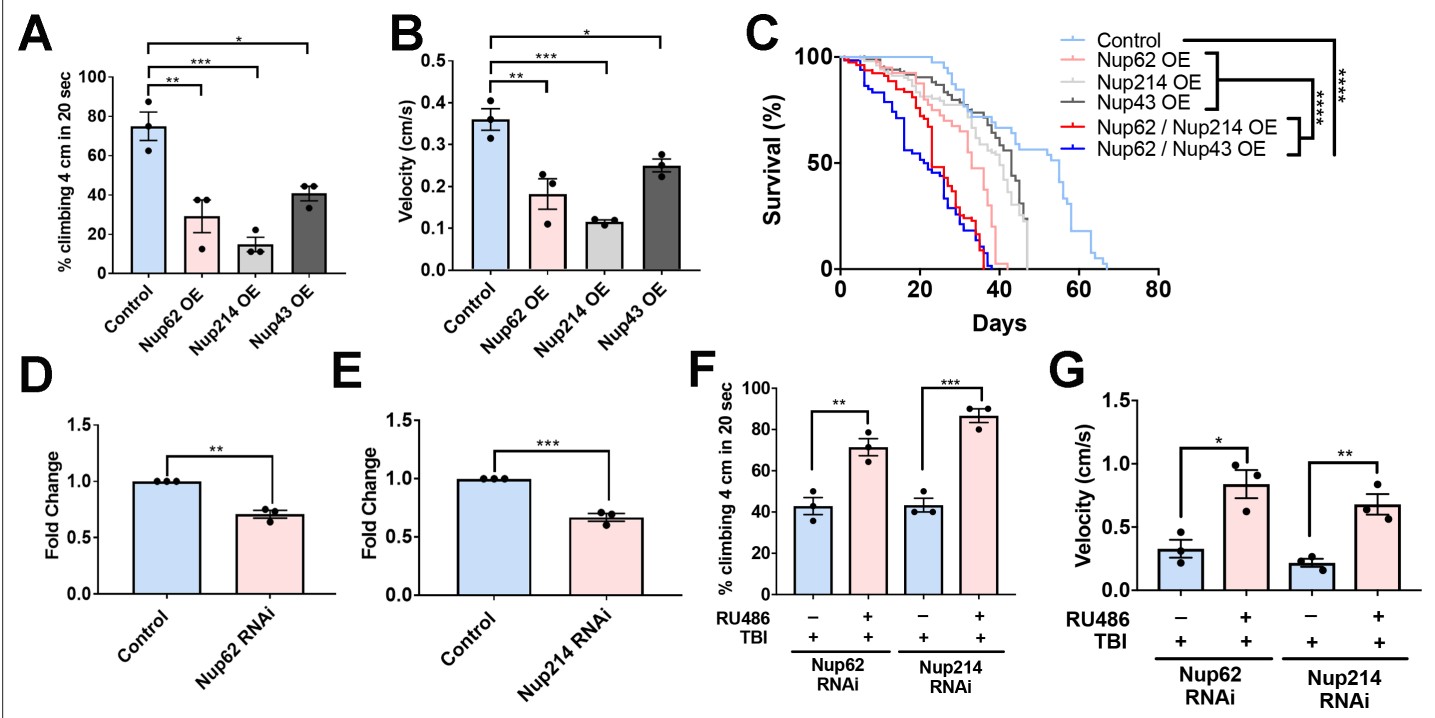

**Figure 5.** Nups expression reduces motor function and lifespan, and knockdown partially rescues trauma-mediated motor dysfunction in *Drosophila*. (**A**) Percentage of flies that climbed 4 cm in 20 seconds for Nup62, Nup214, and Nup43-overexpressing (Nup62 OE, Nup214 OE and Nup43 OE) animals compared to eGFP control (n = 3 trials, 10 animals per trials; ***p < 0.001, *p < 0.01, *p < 0.05). (**B**) Quantification of climbing velocity (cm/s) of Nup62 OE, Nup214 OE, and Nup43 OE flies compared to eGFP controls (n = 3 trials, 10 animals per trials, ***p < 0.001, **p < 0.01, *p < 0.05). (**C**) Kaplan-Meier survival curve of flies expressing Nup62 OE, Nup214 OE, Nup43 OE, Nup62/Nup214 OE, Nup62/Nup43 OE or eGFP control in motor neurons. (n = 60–80, ****p < 0.0001). (**D, E**) qPCR analysis of *Nup62* and *Nup214* mRNA levels in flies expressing Nup62 RNAi, Nup214 RNAi or eGFP controls (n = 3, **p < 0.01, ***p < 0.001). (**F, G**) Adult *Drosophila* expressing Nup62 RNAi or Nup214 RNAi in neuronal cells exposed to repeated traumatic brain injury (TBI) were raised on RU486 (+ RU486) or ethanol (-RU486) treated food for 20 days before motor assays. Quantification of (**F**) percentage of flies that climbed 4 cm in 20 s (n = 3 trials, **p < 0.01, ***p < 0.001) and (**G**) climbing velocity (cm/s) (n = 3 trials, *p < 0.05, **p < 0.01). Log-rank with Grehan-Breslow-Wilcoxon tests were performed to determine significance for panel C, while one-tailed t-tested was used in panel D, E, F, and G. All quantifications represent mean ± s.e.m.

The online version of this article includes the following figure supplement(s) for figure 5:

**Figure supplement 1.** Knockdown of Nup62/Nup214 together extends lifespan.

expression of NUP54 in HEK293T cells did not show endogenous TDP-43 aggregation (*Figure 4—figure supplement 5*), although proteomic analysis showed an increase in *Drosophila* Nup54 protein levels, suggesting that TDP-43 localization and/or aggregation may be influenced by a subset of Nups, specifically NUP62. Expression of human NME1 [*Drosophila* Awd], HSPB2 [*Drosophila* heat shock protein 27 (HSP27)], FKBP1A (*Drosophila* Fkbp12), and SRSF1 (*Drosophila* SF2), which were upregulated as part of the microtubule cytoskeleton, protein folding, and spliceosome component post-TBI (*Supplementary file 1*), in HEK293T cells did not alter TDP-43 localization (*Figure 4—figure supplement 6A,B,C,D*), suggesting that expression of a subset of proteins within the top four upregulated pathways does not disrupt TDP-43 distribution.

## Nucleoporin's expression leads to motor dysfunction and reduced lifespan in *Drosophila* that is partially rescued by nucleoporins knockdown

To determine whether Nup upregulation, which leads to Tbph aggregation and mislocalization, is toxic in *Drosophila*, we assessed the effect of Nup62, Nup214, and Nup43 expression on motor function and lifespan. Overexpression of Nup62, Nup214, or Nup43 in motor neurons significantly reduced motor function and climbing velocity (*Figure 5A and B*, p < 0.01; p < 0.001, and p < 0.05, respectively) in *Drosophila* compared to controls, suggesting that genetic upregulation of Nups alters

motor function. Further, Nup62 OE, Nup214 OE, and Nup43 OE animals had a significantly reduced lifespan compared to control animals—controls had a median survival of 55 days when raised at 29°C, while Nup62 OE, Nup214 OE, and Nup43 animals had a median survival of 33, 40, and 43 days respectively. Overexpression of Nup62 and Nup214 or Nup62 and Nup43 together was further shown to significantly reduce lifespan compared to their respective Nup controls or control animals - Nup62/Nup214 OE and Nup62/Nup43 OE median survival was 23 days and 21 days, respectively (*Figure 5C*, p < 0.0001, *Figure 5—figure supplement 1D*).

Enhanced Nup levels post-TBI coupled with Nup-mediated alteration in motor function in *Drosophila* prompted us to evaluate whether RNAi-mediated knockdown of Nups post-trauma was protective in vivo. We used the Elav-GeneSwitch (ElavGS) system to conditionally express Nup62 RNAi or Nup214 RNAi in neuronal cells by feeding adult flies RU486, as reported (*Anderson et al., 2018*; *Osterwalder et al., 2001*). Adult RNAi (Nup62 or Nup214) flies were exposed to repeated traumatic injuries and motor function as well as lifespan were examined. *Nup62* mRNA (p < 0.01) and *Nup214* mRNA (p < 0.001) levels were significantly reduced in Nup62 and Nup214 RNAi-treated animals compared to controls (*Figure 5D and E*). Interestingly, Nup62 RNAi and Nup214 RNAi animals treated with RU486 post-trauma had a significant improvement in climbing (*Figure 5E*, p < 0.01 and p < 0.001, respectively) and velocity (*Figure 5F*, p < 0.05 and p < 0.01, respectively) compared to ethanol-treated animals, suggesting that genetically modulating *Nup62* and *Nup214* levels post-TBI may partially improve motor function in *Drosophila*. Knockdown of *Nup62* or *Nup214* individually did not significantly extend life-span post-injury (*Figure 5—figure supplement 1A,B*, P = NS). However, RNAi knockdown Nup62/Nup214 together significantly increase lifespan of trauma animals (*Figure 5—figure supplement 1C*,p < 0.001).

## Nuclear pore pathology is present in repetitive TBI patient brain tissues and co-aggregates with TDP-43 pathology

TBI can be categorized as mild, moderate, or severe based on clinical factors such as duration and severity, amnesia and neurological symptoms, and structural brain imaging (Management of Concussion/mTBI *Management of Concussion/mTBI Working Group, 2009*). We examined postmortem brain tissue from patients with mild (nine cases) and severe CTE (nine cases) and aged-matched controls (eight control cases) without neurodegenerative disease for NUP62 pathology in the frontal cortex (*Supplementary file 2*). All mild and severe CTE subjects were involved in sports, specifically American Football with one severe case played Hockey, and control cases played no sports. The mild subjects played an average of 12.222 ± 6.438 years (mean ± SD), while severe subject played an average of 18.333 ± 4.610 (mean ± SD). Interestingly, tissues from mild and severe CTE but not age-matched controls showed widespread and intense NUP62 immunoreactivity (*Figure 6A,B,C* and *Figure 6—figure supplement 1*). Brain tissues from mild and severe CTE cases also showed cytoplasmic and nuclear mislocalization of Nup62 as well as perinuclear accumulation. Cytoplasmic NUP62 aggregates were present in brain tissue from patients with severe CTE, suggesting increased NUP62 aggregation with more severe CTE neuropathology. Enzyme-linked immunosorbent assay (ELISA) on postmortem brain tissues from eleven subjects with mild CTE showed an increasing trend in NUP62 levels, although they were not significantly different from controls cases. However, NUP62 levels in twenty-one severe CTE patient brain tissues were significantly higher than controls (*Figure 6D*, p < 0.05), suggesting that repetitive TBI is associated with altered NUP62 in CTE.

To investigate the connection between Nups and TDP-43 pathology in CTE, we stained the dorsolateral prefrontal cortex at the depth of the cortical sulcus (approximately Broadman area 9) from two subject with mild and severe CTE as well as an aged-matched control. CTE brain tissue exhibited abnormal NUP62 staining in the prefrontal cortex. Interestingly, we found that mild CTE cases showed limited NUP62 coaggregation with pTDP-43, whereas the severe CTE cases showed a more pronounced coaggregation (arrows) of NUP62 and pTDP-43-positive inclusions in the frontal cortex (*Figure 6E*).

## Discussion

Evidence suggests that TBI is a predisposing factor for the development of neurodegenerative diseases (*Chen et al., 2007*; *Gardner and Yaffe, 2015*; *Goldman et al., 2006*; *Lye and Shores, 2000*;

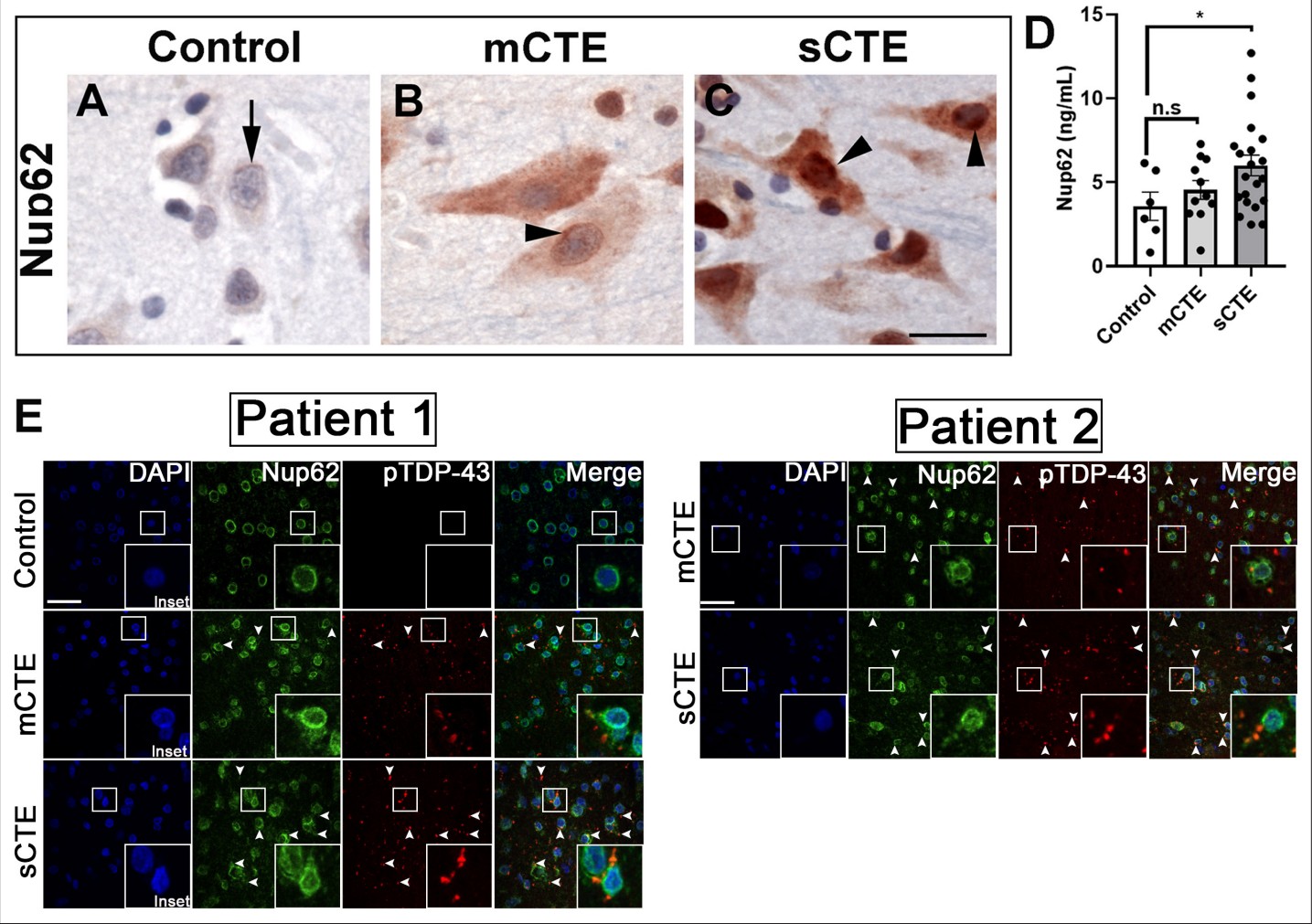

**Figure 6.** NUP62 pathology is present in brain tissue of patients with mild and severe CTE and coaggregates with pTDP-43 inclusion. (**A**) NUP62 immunohistochemical staining in human frontal cortex tissue from a control individual without neurodegenerative disease, showing faint perinuclear staining for NUP62 (arrow). (**B**) NUP62 immunohistochemical staining in human frontal cortex tissue from a participant with mild CTE (mCTE), showing more neurons with diffuse cytoplasmic staining as well as perinuclear aggregates of NUP62 (arrowhead). (**C**) NUP62 immunohistochemical staining in human frontal cortex tissue from a participant with severe CTE (sCTE), showing more intense Nup62 staining in the nucleus and cytoplasm (arrowheads). Scale bar = 20 μm. (**D**) Quantification of enzyme-linked immunosorbent assay (ELISA) results for human NUP62 in control (n = 6), mCTE (n = 11), and sCTE (n = 21) subjects (*p < 0.05, n.s. = not significant). (**E**) Representative immunofluorescence images of a control, and two mCTE and sCTE cases stained for NUP62 (green) and phosphorylated TDP-43 (red) in the frontal cortex (Scale bar = 25 μm), showing pTDP-43 co-localization with NUP62 (arrows head, merge) in two patients. DAPI was used as a nuclear marker. One-way ANOVA with Tukey's multiple comparisons tested was used for panel D. All quantification represent mean ± s.e.m.

The online version of this article includes the following figure supplement(s) for figure 6:

**Figure supplement 1.** Control NUP62 immunohistochemistry cases.

Peters et al., 2013; Sivanandam and Thakur, 2012; Taylor et al., 2016; Vanacore et al., 2006), and repeated trauma to the head is also associated with CTE (McKee et al., 2013; Omalu et al., 2010; Omalu et al., 2005; Stern et al., 2011). Further, CTE postmortem brain tissues and animal models of TBI show TDP-43 and TAU pathology (Blennow et al., 2012; Huang et al., 2017; Mckee et al., 2018; McKee et al., 2010; Tan et al., 2018; Wang et al., 2015),, although the underlying mechanism that promotes this pathology has remained unexplored. Interestingly, although TBI causes nuclear dysfunction (Tajiri et al., 2016), it has remained unknown whether TBI leads to NCT defects that might be linked to TDP-43 pathology in CTE. Here, we uncovered that TBI disrupts NPC proteins and NCT in models of trauma, postmortem brain tissues form CTE patients, and leads to TDP-43/Tbph coaggregation with components of the NPC. Of note, our study provides evidence of alteration in

nucleoporins proteins, defects in NPC morphology and RanGAP1 distributions, coaggregation of NPC components with TDP-43/Tbph, overexpression of nucleoporins lead to TDP-43 mislocalization and/or aggregation, alter motor function and lifespan, and pharmacological inhibition of nuclear export or modulating nucleoporins post-injury is protective. Taken together, these results suggest that TBI can disrupt NCT machinery. These defects may account for the nuclear depletion and cytoplasmic accumulation of TDP-43 widely seen in CTE and/or TBI-related conditions.

Trafficking of protein and RNA between the nucleus and cytoplasm is important for proper cellular function—NCT of macromolecules is essential for signaling, neuronal plasticity, and overall cellular survival (*Dickmanns et al., 2015*; *Schachtrup et al., 2015*). NCT dysfunction has been reported in AD (*Eftekharzadeh et al., 2018*), HD (*Gasset-Rosa et al., 2017*; *Grima et al., 2017*), and ALS/FTD (*Chou et al., 2018*; *Zhang et al., 2015*), and mutation in NUP62 causes autosomal recessive infantile bilateral striatal necrosis (*Basel-Vanagaite et al., 2006*), suggesting a common disrupted pathway. Our study provides evidence that TBI directly alters NPC and NCT components in vivo. Nups mRNA and protein levels were increased post-TBI (*Figure 1*) and NUP62 protein levels were elevated in post-mortem brain tissues from CTE patients (*Figure 6*). Nups that were significantly upregulated include cytoplasmic FG Nups and filaments (Nup214), central channel FG Nups (Nup62 and Nup54), nuclear FG Nups and basket (TPR), linker Nups (Nup93), and outer ring Nups (Nup44A/SEH1, Nup43, and Nup75). In mammals, NUP93 forms an adaptor complex with NUP155, NUP35, NUP205, and NUP188, while NUP214 and other Nups constitute NPC cytoplasmic filaments that provide key interaction sites for NCT machinery (*Lin and Hoelz, 2019*; *Solmaz et al., 2011*). On the other hand, NUP62 and NUP54, in addition to NUP58, are suggested to constitute the channel (termed channel Nups) that undergoes large-scale reversible expansions to regulate NCT (*Dickmanns et al., 2015*; *Lin and Hoelz, 2019*) suggesting that alteration in these components by TBI might impact NCT of critical components in and out of the nucleus.

It is tempting to speculate that upregulation in Nups protein post-trauma (*Figure 1*) may be linked to post-transcriptional modifications. NPC components are extensively modified by nucleocytoplasmically disposed *N*-acetyglucosamine (GlcNAc) residues *O*-linked to serine and threonine residues (*O*-GlcNAc) (*Holt et al., 1987*; *Li and Kohler, 2014*). Indeed, 18 of the 30 Nups are modified by *O*-GlcNAc, and FG-Nups in the central scaffold of the NPC are heavily *O*-GlcNAcylated (*Zhu et al., 2016*). Nups such as Nup62, Nup153, Nup214, Nup93, and Nup358 as well as scaffold Nups are targeted by *O*-GlcNAc (*Boyce et al., 2011*; *Gloster et al., 2011*; *Zhu et al., 2016*). Further, genetic and pharmacological reduction of *O*-GlcNAc levels lead to decreased Nup levels and accelerated degradation, suggesting that *O*-GlcNAc might modulate Nup stability (*Zhu et al., 2016*). Collectively, these studies imply that repeated trauma can directly or indirectly upregulate components of the NPC and that *O*-GlcNAcylation may have an important role in this process. However, further studies are needed to assess the role of trauma in modulating Nup *O*-GlcNacylation. Alternatively, Nups protein upregulation post-trauma may occur at the transcriptional level. Our results showed significant upregulation in Nups mRNA (*Figure 1*). It is plausible that both transcriptional and translational events may contribute to Nups upregulation in traumatic injury, but the exact sequence of events and factors involve require additional work. Intriguingly, Nups such as Pom121 and Nup107 are long-lived proteins in post-mitotic cells and are susceptible to oxidative damage. The slow turnover of damaged Nups and aged-associated reduction of nuclear protein import and NPC may result in neuronal dysfunction and death (*D'Angelo et al., 2009*; *Pujol et al., 2002*). In support, our data showed the Nup214 protein levels remains elevated overtime in larval and adult brains. It is possible that the slow turnover of Nup214 protein might be due to dysfunction in protein clearance pathways. Our lab and others have shown that traumatic brain injury impaired autophagy (*Anderson et al., 2018*; *Sarkar et al., 2014*; *Zeng et al., 2018*) and ubiquitin proteosome components (*Tylicka et al., 2014*; *Yao et al., 2008*), suggesting that the impairments in these pathways might be important in mediating Nups slow turnover which potentially contribute to neurodegeneration.

Although strong evidence exists for NCT dysfunction in neurodegenerative disease, this evidence is centered around how pathological mutations in disease proteins influence this process, while many neurodegenerative diseases are mostly sporadic. Factors such as heavy metal exposure, pesticides, smoking, and head trauma have been link to development of neurodegenerative disease. Repeated head trauma is one of the most consistent risk factors for developing diseases such as ALS (*Peters et al., 2013*) and AD (*Fann et al., 2018*). However, the role of repeated head trauma in NCT dysfunction and

its relation to neurodegeneration are unknown. To address this question, we used a well-characterized *Drosophila* trauma model (*Katzenberger et al., 2015*; *Katzenberger et al., 2013*) that shows a strong correlation between the amount of trauma and lethality as well as pathology in larval and adult *Drosophila* (*Anderson et al., 2018*). Our results show that repeated trauma altered several molecular pathways including NPC and NCT. The NPC tightly regulates transport in and out of the nucleus, suggesting that trauma-mediated alteration in NPC components may disrupt NCT. Indeed, trauma disrupt the NPC morphology resulting in gaps and/or aggregation of nucleoporins on the membrane (*Figure 2*). Trauma was shown to disrupt RanGAP1 distribution, Ran-dependent nuclear import and export regulator (*Bischoff et al., 1995*), as well as nuclear import of GFP containing a classic NLS and NES in vivo (*Figure 3*). Studies in multiple model systems of neurodegenerative diseases suggest that defects in the NCT pathway can manifest as 'leaky pores', impaired NCT, reduced NPC density, and irregular nuclear envelope morphology (*Hutten and Dormann, 2020*). Studies also show that over-expression of NUP214 leads to NCT defects (*Boer et al., 1998*). Conversely, depletion of NUP214 in mouse embryonic stem cells is reported to impair NLS-mediated protein import and lead to nuclear accumulation of polyadenylated RNA (*van Deursen et al., 1996*), while knockout of NUP98 in mice disrupts nuclear import of proteins with an NLS (*Wu et al., 2002*). Together, these data implicate repeated traumatic injury as a disruptor of the NPC and suggest that altering nucleoporins might possibly disrupt NCT by either disrupting the NPC morphology or creating 'leaky pores' but further work is needed to address this.

Our data further support a role for Nup62 upregulation in transport defects of NLS-containing protein, TDP-43. Targeted upregulation of Nup62 in *Drosophila* and in HEK293T cells caused nuclear depletion of TDP-43. In addition, expression Nup214, Nup43, and Nup93 also caused TDP-43 mislocalization, albeit to a lesser extent. Our in vivo and in vitro data further support that Nup62 upregulation leads to TDP-43 cytoplasmic aggregation and phosphorylation as well as increased levels of insoluble TDP-43. TDP-43 contains a classic NLS sequence, and its aggregation and phosphorylation are reported to be toxic (*Johnson et al., 2009*; *Zhang et al., 2009*). TDP-43 is considered a pathological hallmark of ALS/FTD (*Lattante et al., 2013*) and is also found in CTE postmortem brain tissues (*McKee et al., 2010*). Further, our results show that mild and severe CTE cases also have NUP62 pathology that coaggregated with pTDP-43, and its extent is related to disease severity. Importantly, repeated trauma in *Drosophila* led to TDP-43/Tbph accumulation with NPC components in the brain and this is recapitulated by overexpressing Nup62 but not Nup214 or Nup43 in *Drosophila*. We speculate that the TDP-43 pathology observed post-trauma is likely due to NPC dysfunction via upregulated NPC proteins. Our recent work have demonstrated that cytoplasmic Nup62 droplets exhibit characteristics of proteins that undergo liquid-liquid phase separation, and it is likely that these interaction through the classical NLS promotes deleterious phase transition of cytoplasmic TDP-43 causing it to mature into insoluble inclusions (*Gleixner et al., 2021* **PREPRINT**), providing a mechanism for how Nups might promote TDP-43 pathology in traumatic injury. On the other hand, evidence in vitro and in vivo demonstrate that TDP-43 aggregation/demixing promotes NPC deficits that sequesters NPC components with TDP-43 aggregates (*Chou et al., 2018*; *Cook et al., 2020*; *Gasset-Rosa et al., 2019*; *Zhang et al., 2018*), suggesting that the TBI-mediated NCT deficits could be a consequence of cytoplasmic TDP-43 pathology. However, further research is needed to address whether NPC dysfunction is a cause or consequence of TDP-43 pathology in trauma-related conditions to fully decipher the exact mechanism. Moreover, the synergistic effects of both mechanisms cannot be ruled out as cause of neurodegeneration in traumatic injury.

Here, we also show that overexpression of Nups is toxic in *Drosophila*. Overexpression of Nup62, Nup214, or Nup43 reduces motor function and lifespan of *Drosophila* (*Figure 5*), suggesting that altered NPC proteins may mediate toxicity. Interestingly, RNAi-mediated knockdown of *Nup62* or *Nup214* partially rescued the motor dysfunction but not lifespan in adult *Drosophila* exposed to TBI. However, knockdown of Nup62 and Nup214 together post-trauma partially extend lifespan. Importantly, pharmacological inhibition of nuclear export via administration of KPT compounds rescued the trauma-mediated NCT defect of GFP-tagged NLS-NES and Tbph aggregation, which correlates with rescue of trauma-mediated lethality (*Figure 3*). Similarly, KPT-350 and KPT-276 are protective in models of ALS/FTD (*Tamir et al., 2017*; *Zhang et al., 2015*) and TBI (*Cantu et al., 2016*; *Tajiri et al., 2016*), suggesting a potential therapeutic application in TBI cases. While we did not observe a complete rescue of lifespan by RNAi knockdown of Nups post-injury, we speculate that it might be

due to upregulation of multiple Nups post-trauma that might contribute to the overall phenotypes. We cannot exclude the possibility that Nups upregulation or a combination of other upregulated Nups might mediate toxicity by altering NCT balance, leading to NLS-containing proteins such as TDP-43 mislocalization and aggregation. Thus, further work is needed to assess whether modulating the levels of multiple Nups post-trauma or in conjunction with NCT inhibitors might be protective. In all cases, modulating Nups and/or nuclear export post-injury might provide some level of protection following repeated head trauma.

Taken together, our data suggest that disruption of NPC and NCT function may be a common pathogenesis in repetitive TBI diseases such as CTE. Based on our findings, we propose that trauma-mediated cytoplasmic aggregation of NLS-containing proteins such as TDP-43 is potentially caused by NCT defects and that cytoplasmic aggregation of NPC components such as Nup62 may sequester TDP-43 in aggregates that lead to toxicity. Further understanding of the role of repeated trauma in NCT and TDP-43 pathology will help develop therapeutic strategies for CTE and trauma-related diseases.

# Materials and methods

## Key resources table

| Reagent type (species) or resource | Designation | Source or reference | Identifiers | Additional information |
|---|---|---|---|---|
| Genetic reagent (*D. melanogaster*) | w[1118] | Bloomington *Drosophila* Stock Center | BDSC:3605; FLYB: FBst000360; RRID:BDSC_3605 | w[1,118] |
| Genetic reagent (*D. melanogaster*) | w[1118]; P{UAS-NLS-NES[+]-GFP}5A | Bloomington *Drosophila* Stock Center | BDSC:7032; FLYB: FBst0007032; RRID:BDSC_7032 | w[1,118]; P{w[+ mC] = UAS-NLS-NES[+]-GFP}5A |
| Genetic reagent (*D. melanogaster*) | y[1] w*; P{UAS-NLS-NES[P12]-GFP}2A | Bloomington *Drosophila* Stock Center | BDSC:7033; FLYB: FBst0007033; RRID:BDSC_7033 | y (**Amador-Ortiz et al., 2007**) w[*]; P{w[+ mC] = UAS-NLS-NES[P12]-GFP}2A |
| Genetic reagent (*D. melanogaster*) | w[1118]; P{EP}Nup44A[EP2417] | Bloomington *Drosophila* Stock Center | BDSC:17053; FLYB: FBst0017053; RRID:BDSC_17053 | w[1,118]; P{w[+ mC] = EP}Nup44A[EP2417] |
| Genetic reagent (*D. melanogaster*) | w[1118]; P{EPg}Nup93-2[HP35056] | Bloomington *Drosophila* Stock Center | BDSC:21975; FLYB: FBst0021975; RRID:BDSC_21975 | w[1,118]; P{w[+ mC] = EPg}Nup93-2[HP35056] |
| Genetic reagent (*D. melanogaster*) | P{VSH330104}attP40 | Vienna *Drosophila* Resource Center | VDRC:v330104; FLYB: FBst0491076; RRID:FlyBase_FBst0491076 | P{VSH330104}attP40 |
| Genetic reagent (*D. melanogaster*) | P{KK108318}VIE-260B | Vienna *Drosophila* Resource Center | VDRC: v100588; FLYB: FBst0472461; RRID:FlyBase_FBst0472461 | P{KK108318}VIE-260B |
| Genetic reagent (*D. melanogaster*) | UAS-Nup62 | This paper | | Created at BestGene |
| Genetic reagent (*D. melanogaster*) | M{UAS-Nup214.ORF}ZH-86Fb | FLYORF | FLYORF: F001467; FLYB: FBst0500182; RRID:FlyBase_FBst0500182 | M{UAS-Nup214.ORF}ZH-86Fb |
| Genetic reagent (*D. melanogaster*) | M{UAS-Nup43.ORF.3xHA.GW}ZH-86Fb | FLYORF | FLYORF: F003133; FLYB: FBst0502466; RRID:FlyBase_FBst050246 | M{UAS-Nup43.ORF.3xHA.GW}ZH-86Fb |
| Genetic reagent (*D. melanogaster*) | CRISPR/Cas9 hTDP43-WT | Gift from David B. Morton **Chang and Morton, 2017** | | |
| Cell line (*Homo sapiens*) | Human Embryonic Kidney cells (HEK 293T) | ATCC | RRID:CVCL_0063 | |
| Transfected construct (human) | mRFP-FKBP1A Plasmid | Addgene | (Cat #: 67514); RRID:Addgene_67514 | transfected construct (human) |
| Transfected construct (human) | Frt-V5-HspB2 Plasmid | Addgene | (Cat #: 63103); RRID:Addgene_63103 | transfected construct (human) |
| Transfected construct (human) | FLAG NM23-H1/NME1 Plasmid | Addgene | (Cat #: 25000); RRID:Addgene_25000 | transfected construct (human) |
| Transfected construct (human) | pEGFP SF2/SRSF1 Plasmid | Addgene | (Cat #: 17990); RRID:Addgene_17990 | transfected construct (human) |
| Transfected construct (human) | Nup54-HA-eGFP Plasmid | VectorBuilder | This paper | transfected construct (human) |

*Continued on next page*

*Continued*

| Reagent type (species) or resource | Designation | Source or reference | Identifiers | Additional information |
|---|---|---|---|---|
| Transfected construct (human) | HA-eGFP Plasmid | VectorBuilder | This paper | transfected construct (human) |
| Transfected construct (human) | mRuby Plasmid | Gift from Dr. Christopher Donnelly | This paper | transfected construct (human) |
| Transfected construct (human) | NUP62-mRuby Plasmid | Gift from Dr. Christopher Donnelly | This paper | transfected construct (human) |

| Reagent type (species) or resource | Designation | Source or reference | Identifiers | Additional information |
|---|---|---|---|---|
| Antibody | anti-Nup214 (Guinea Pig polyclonal) | Gift from Dr. Christos Samakovlis *Roth et al., 2003* | | WB (1:5000) |
| Antibody | Anti-Lamin Dm0 (Mouse monoclonal) | DSHB | Cat# ADL84.12; RRID:AB_528338 | IF(1:200) WB (1:1000) |
| Antibody | anti-α-Tubulin (Mouse monoclonal) | Sigma-Aldrich | Cat#: T5168; RRID:AB_477579 | WB (1:10,000) |
| Antibody | anti-Tbph (Rabbit polyclonal) | Gift from Dr. Frank Hirth *Diaper et al., 2013* | | IF (1:1500) WB (1:3000) |
| Antibody | anti-FLAG (Mouse monoclonal) | Sigma-Aldrich | Cat#: F1804; RRID:AB_259529 | IF (1:1000) |
| Antibody | anti-GFP (Chicken polyclonal) | Abcam | Cat#: ab13970; RRID:AB_300798 | IF (1:1000) |
| Antibody | anti-TDP43 (Rabbit polyclonal) | Proteintech | Cat#: 10782–2-AP; RRID:AB_615042 | IF (1:1000) |
| Antibody | anti-phospho-TDP43 (Rat monoclonal) | Millipore SIGMA | Cat#: MABN14; RRID:AB_11212279 | IF (1:1000) |
| Antibody | anti-Mab414 (Mouse monoclonal) | Abcam | Cat#: ab24609; RRID:AB_448181 | IF (1:1000) |
| Antibody | anti-RanGAP1 (Rabbit polyclonal) | Millipore SIGMA | Cat#: ABN1674 | IF (1:200) |
| Antibody | anti-Nup62 (Mouse polyclonal) | BD Transduction Laboratories | Cat#: 610497; RRID:AB_397863 | IHC/IF (1:500) |
| Antibody | anti-pTDP-43 (Rabbit polyclonal) | Cosmo Bio | Cat#:NC0877946; RRID:AB_1961899 | IF (1:1000) |
| Antibody | anti-Nup62 (Mouse polyclonal) | Roche Applied Science | Cat#: 610497; RRID:AB_397863 | IHC (1:400) |
| Antibody | anti-RanGAP1 (Rabbit polyclonal) | Santa Cruz Biotechnology | Cat#: sc-28322; RRID:AB_2176987 | IHC (1:1000) |
| Sequence-based reagent | *Futsch* | This paper | PCR primers (5'–3'): | Forward: CAAAGCCCACTCACCTTTC Reverse: CTGCTCCTGCCAACATCT Probe: AGTCTCTGGAAATGCAGCACCACT |
| Sequenced-based reagent | *dNup93-2* | This paper | PCR primers (5'–3'): | Forward: ACACCGTCCGCGAAATAC Reverse: ACTCAACCGCCACCTTAAC Probe: ATGGCCGCTGGTTTACTACGGATT |
| Sequenced-based reagent | *dNup214* | This paper | PCR primers (5'–3'): | Forward: CCTAAGTGAGGACAAGGATGAG Reverse: GGCATAGTCTGCAGCTTCTT Probe: TGCCTTCGACACTTCTACAACGCA |
| Sequenced-based reagent | *dNup54* | This paper | PCR primers (5'–3'): | Forward: GAGTGAGCTGACAGAACTCAAG Reverse: CTCGGCCAGTTTCCGTTTAT Probe: CCACTGCCACAGCGAAGATACTTGA |
| Sequenced-based reagent | *dNup44A* | This paper | PCR primers (5'–3'): | Forward: AAGGTATCCTCCACCAATACCC Reverse: TTGGGTGCAAACTCCACATC Probe: TTGTAGACTCGCGGACCAGTGTCA |
| Sequenced-based reagent | *dNup62* | This paper | PCR primers (5'–3'): | Forward: CTTGCTGTTGTCTGCATCTC Reverse: CAGCACCAGCTTCAGGA Probe: ACATTCTCTTTCGGAACACCGGCA |
| Sequenced-based reagent | *Emb (Exportin)* | This paper | PCR primers (5'–3'): | Forward: GGTCACGCGTATGTCATTCA Reverse: GTTCACATTGACGCCATTCAC Probe: AGCATGTCCAGATATATGCGGCCC |
| Sequenced-based reagent | *dNup43* | This paper | PCR primers (5'–3'): | Forward: TTGCATCTGATCCTCCTCCAC Reverse: ACCGCCATGGAGTTCGT Probe: TGGACGTTAAACACGCTGAGATGACC |

*Continued*

| Reagent type (species) or resource | Designation | Source or reference | Identifiers | Additional information |
|---|---|---|---|---|
| Sequenced-based reagent | *dGapdh* | This paper | PCR primers (5'–3'): | Forward: CAACAGTGATTCCCGACCAG<br>Reverse: TTCGTCAAGCTAATCTCGTGG<br>Probe: CCAAAACTATCGTACAAACCCGGCG |
| Commercial assay or kit | 3,3'-diaminobenzidine (DAB) | Vector Laboratories | Cat: #SK-4100; RRID:AB_2336382 | |
| Commercial assay or kit | NE-PER nuclear-cytoplasmic extraction kit | ThermoFisher Scientific | Cat #: 78,833 | |
| Commercial assay or kit | Nup62 ELISA | LSBio | Cat #: LS-F22196 | |
| Chemical compound, drug | KPT-350 | Karyopharm | | |
| Chemical compound, drug | KPT-276 | Selleck Chem | Cat #: S7251 | |
| Software, algorithm | GraphPad Prism 6 | GraphPad Prism 6 | RRID:SCR_002798 | |

## *Drosophila*

### Fly lines

w[1118], P{EP}Nup44A[EP2417], P{EPg}Nup93-2[HP35056], P{UAS-NLS-NES-GFP}, P{UAS-NLS-ΔNES-GFP}, Ok371-gal4, and UAS-eGFP were obtained from the Bloomington *Drosophila* stock center (http://www.flystocks.bio.indiana.edu). UAS-Nup214 (F001467) and UAS-Nup43 (F003133) were obtained from FlyORF. Nup62 RNAi (100588) and Nup214 *RNAi* (41964) were obtained from Vienna *Drosophila* Resource Center. The previously described CRISPR/Cas9 human TDP-43 wild-type (hTDP-43WT) was a kind gift from David B. Morton (Oregon Health and Science University) (*Chang and Morton, 2017*). UAS-Nup62 overexpression lines (UAS-Nup62 OE) were generated through site-specific integration of the transgene at BestGene Inc To generate UAS-Nup62 OE lines, we used the pUAST-attB vector. We subcloned the Nup62 cDNA (*Drosophila* Genomic Research Center) into the pUAST-attB vector.

### Proteomics of *Drosophila* brains

Larvae (three biological replicate) that were subjected to repeated whole-body trauma, 8 hits @ 50° angles with 20 s interval between hits (*Anderson et al., 2018*), and non-TBI controls brains were extracted immediately following trauma and flash-frozen. For protein extraction, frozen brains were thawed on ice with 50 μL of chilled MS lysis buffer (6 M urea, 0.4 M NaCl, 20 mM HEPES [pH 7.9], 5 % glycerol, 0.2 mM EDTA, 0.5 mM DTT) with 1 X protease inhibitors. Brains were crushed by 20 strokes with a plastic pestle and centrifuged at 19,000 *g* for 30 min to pellet debris. Supernatants was passed through a 0.45 μm microcentrifuge filter and quantified for mass spectrometry by BCA assay. Filtered protein was reduced and alkylated with 10 mM TCEP and 40 mM chloroacetamide, respectively. Lysates were digested into tryptic peptides using the filter-aided sample prep (FASP) method (*Wisniewski et al., 2009*) by diluting to 4 M urea, adding LysC (Wako), and rocking at ambient temperature for 4 hr. Lysates were further diluted to ~1 M urea for trypsin digestion (Pierce), rocking overnight at ambient temperature. Digested peptides were desalted using an Oasis HLB cartridge (Waters) according to the manufacturer's instructions.

Samples were suspended in 7 μL of 3 % (v/v) acetonitrile/0.1 % (v/v) trifluoroacetic acid, and 1 μL was directly injected on a C18 1.7 μm, 130 Å, 75 μm X 250 mm M-class column (Waters), using a Waters M-class UPLC. Tryptic peptides were eluted at 300 nL/min using a gradient from 3–20% acetonitrile over 100 min into an Orbitrap Fusion mass spectrometer (Thermo Scientific). Precursor mass spectra (MS1) were acquired at a resolution of 120,000 from 380 to 1500 m/z with an AGC target of $4.0 \times 10^5$ and a maximum injection time of 60 ms. Dynamics exclusion was set for 30 s with a mass tolerance of ±10 ppm. Precursor peptide ion isolation width for MS2 fragment scans was 1.6 Da using the quadrupole, and the most intense ions were sequenced using Top Speed with a 3 s cycle time. All MS2 sequencing was performed using higher energy collision dissociation (HCD) at 35 % collision energy and scanning in the linear ion trap. An AGC target of $1.0 \times 10^4$ and 35 s maximum injection time was used. Raw files were searched against the Uniprot *Drosophila* database using Maxquant version 1.6.0.16 with cysteine carbamidomethylation as a fixed modification (*Bateman, 2019Old et al., 2005*;

*Tyanova et al., 2015*). Methionine oxidation and protein N-terminal acetylation were searched as variable modifications. All peptides and proteins were thresholded at a 1% false discovery rate (FDR).

## Quantification of AE-MS data and gene ontology

Of the 4696 *Drosophila* proteins detected, the top 2000 ranking proteins were selected for further analysis with >4 label-free quantified (LFQ) values averaged across all samples and biological replicates. The average fold-change between TBI samples and controls (n = 2 or 6 biological replicates, respectively) was determined, and significance was determined by student's t-test. Proteins were selected for GO analysis if they had a log-2 fold-change after TBI of ≥0.7 or ≤–0.7. The 361 proteins are just those whose p-value between TBI and non-TBI is ≤0.05. When we did both >1.6 fold up or down and p ≤ 0.05, there were 246 proteins (*Supplementary files 1 and 3*) that were altered and present "TBI Proteome". The GO analysis accepted all genes changed >1.6 fold and p ≤ 0.1 to get enough genes to do GO analysis on. These proteins were consistently observed in all replicates. GO analysis was performed using Cytoscape v.3.7.1 and the BiNGO tool set v.3.0.3 (*Saito et al., 2012*). Associated GO terms were determined with the hypergeometric test with Benjamini and Hochberg FDR correction. Volcano plots were generated using all proteins associated with the selected GO terms and in the top 2000 ranked proteins detected. Additional proteins were included in volcano plots by manual curation and indicated in recent literature to associate with factors listed under the GO term (e.g. nuclear pore proteins).

## Western blotting

Fly heads were collected from each genetic cross and snap-frozen on dry ice. Heads were crushed on dry ice and incubated in RIPA buffer (150 mM NaCl, 1% NP40, 0.1% SDS, 1% sodium deoxycholate, 50 mM NaF, 2 mM EDTA, 2 mM DTT, 0.2 mM Na orthovanadate, 1 X Roche protease inhibitor #11836170001). Lysates were sonicated and centrifuged to remove debris. Supernatants were boiled in Laemmli buffer (Boston Bioproducts, #BP-111R) for 5 min and loaded onto 4%–12% Nupage Bis-Tris gels (Novex/Life Technologies). Proteins were transferred using the iBlot2 (Life Technologies, #13120134) onto nitrocellulose (iBlot 2 NC regular Stacks, Invitrogen, #IB23001). Western blots were blocked with milk solution (BLOT-QuickBlocker reagent, EMB Millipore, #WB57-175GM) and incubated with primary antibody overnight: guinea pig DNup214, 1:5000 (a kind gift from Christos Samakovlis [*Roth et al., 2003*]); anti-lamin Dm0 (ADL84.12), 1:1000 (Developmental Studies Hybridoma Bank); mouse anti-tubulin, 1:10,000 (Sigma-Aldrich); rabbit GAPDH, 1:1000 (GeneTex). Blots were washed and incubated in secondary antibody for one hour [anti-mouse IRDye 680D, 1:10,000 (LI-COR Biosciences); anti-rabbit DYLight 800, 1:10,000 (Invitrogen)] and imaged on a Licor imager (Odyssey CLx). All western blots were run in triplicate using biological replicates. Protein levels were quantified using NIH Image J software, and statistical analysis was performed with GraphPad Prism six software.

## Soluble-insoluble fractionation

Ten adult fly heads were collected for each genotype, crushed on dry ice, incubated in NP40 lysis buffer (10 mM Tris HCl pH 7.8, 10 mM EDTA, 150 mM NaCl, protease inhibitor cocktail, 0.5% NP40), sonicated, and centrifuged (21,000 *g* for 20 min), and supernatants (soluble fractions) were collected. Pellets were washed in 1 mL washing buffer (50 mM Tris HCl pH 7.4, 150 mM NaCl), vortexed, and centrifuged (21,000 *g* for 5 min). Supernatants were discarded, and pellets were resuspended in resolubilization buffer (50 mM Tris HCl pH 6.8, 5% SDS, 10% glycerol) and sonicated. Similarly, transfected HEK293T cells were harvested 24 h after transfection, and the above protocol was followed. Western blotting was performed as described above.

## Nuclear-cytoplasmic fractionation

A total of 50 adult fly heads were harvested, and nuclear-cytoplasmic fractionation was done using a NE-PER Nuclear-Cytoplasmic extraction kit per manufacturer's protocol (Thermo Fisher Scientific).

## Larval preparations, immunohistochemistry, and quantification

Third instar larvae or adult *Drosophila* brains were dissected, fixed, and immunostained as previously described (*Anderson et al., 2018*). Briefly, animals were dissected in ice-cold phosphate buffered saline (PBS) (Lonza, #17–516 F), fixed in 4% formaldehyde, washed three times in PBS, incubated

in 5% Triton X-100/PBS for 20 min, washed three times in 0.1% PBST (0.1 % Triton X-100/PBS), and incubated overnight with primary antibodies: mouse anti-Lamin Dm0, 1:200 (Developmental Studies Hybridoma Bank); rabbit anti-Tbph, 1:3000 generous gift from Dr. Frank Hirth *Diaper et al., 2013*; mouse anti-FLAG, 1:1,000 (Sigma-Aldrich); chicken anti-GFP, 1:1000 (Abcam); Rabbit Anti-RanGAP1, 1:200 (Millipore Sigma) or mouse anti-nuclear pore complex (Mab414), 1:1000 (Abcam). Larvae were washed three times in 0.1% PBST and incubated with secondary antibodies: anti-rabbit Alexa Flour 568, 1:100 (Invitrogen, #651727); anti-mouse Alexa Flour 647, 1:100 (Invitrogen, #28181); anti-rabbit Alexa Flour 405, 1:100 (Invitrogen, #157554); anti-rabbit Alexa Flour 647, 1:100 (Invitrogen, #1660844); or anti-chicken Alexa Flour 647, 1:100 (Invitrogen, #2010133). Stained larvae were mounted using DAPI Fluoroshield (Sigma-Aldrich, #F6182). Images were collected on a Nikon A1 eclipse $T_i$ confocal microscope. Relative Tbph puncta and staining intensity were quantified in 100–150 VNC cells from four to seven animals using the Image J threshold function, and average intensity (subtracted from background) was used for analysis. All IF image quantification were done single-blinded.

## Eclosion assay

For eclosion percentages, a total of 15 $w^{1118}$ larvae were subjected to trauma, 8hits @ 50° angles with 20 sec interval between hits, as previously described (*Anderson et al., 2018*). Briefly, larvae were exposed to trauma and directly transferred to food vials supplemented with DMSO only, KPT-350 (0.05, 0.1, or 0.5 mM; Karyopharm), or KPT-276 (50, 200, or 1,000 nM; Selleck Chem). Eclosion percentages were calculated as: (total number of eclosed adults) / (total number of pupal cases) x 100. Each condition was analyzed in triplicate using biological replicates. Statistical analysis was performed with GraphPad Prism six software.

## Climbing assay

OK371-gal4 (motor neuron specific driver) was used to overexpress Nups (Nup62 OE, Nup214 OE, Nup43 OE) or eGFP (control), and climbing assays were performed as previously described (*Anderson et al., 2018*). Additionally, Elav-GeneSwitch (ElavGS) was used to conditionally express *Nup62* RNAi or *Nup214* RNAi in neuronal cells on normal cornmeal food. Adult progeny were subjected to repeated trauma as described (*Anderson et al., 2018*), and flies that survived after 24 hr were placed on food treated with ethanol (-RU486) or RU486 (+ RU486, Cayman Chemical) for 20 days before motor assay. Adult flies were anesthetized with $CO_2$ after 20 days, transferred to vials, and allowed to recover for 30 min. Flies were knocked to the bottom of vials by tapping against the bench three times, and a video camera was used to record flies climbing up the walls. The percentage of flies that climbed 4 cm in 20 s as well as the velocity (cm/s) of each individual fly was quantified and analyzed using GraphPad Prism six software. Three experimental replicates were performed for each group.

## Lifespan assay

Nup62 OE, Nup214 RNAi or Nup43 RNAi or eGFP adult 2-day-old flies (20 flies per vial, 60–80 flies) were collected for each experimental group. Flies were transferred to fresh food twice a week. Number of dead flies was counted every day, and survival functions were calculated and plotted as Kaplan-Meier survival curves. Log-rank with Grehan-Breslow-Wilcoxon tests were performed to determine significance of differences in survival data between samples using GraphPad Prism six software.

## *Quantitative reverse-transcriptase polymerase chain reaction (qRT-PCR)*

RNA was extracted in triplicate from TBI or non-TBI $w^{1118}$ larval brains (10 brains per extraction) as well as Nup62 RNAi, Nup214 RNAi, Nup62 RNAi, eGFP, Nup214 OE, Nup43 OE, or Nup62 OE, flies using Trizol (Ambion, #15596026) in 1-bromo-3-chloropropane (BCP; Sigma-Aldrich, #MKCB0830V), as previously described (*Jensen et al., 2013*). Brains were dissected from larvae, snap frozen on dry ice, ground in Trizol, and centrifuged to pellet. BCP was added, and samples were centrifuged again. The upper aqueous layer was used to precipitate RNA with isopropanol, followed by pelleting RNA through centrifugation. RNA pellets were washed with 70% ethanol and allowed to air-dry prior to resuspending in RNase-free water. RNA samples were quantified on a NanoDrop ND-1000 spectrophotometer, and purity was assessed using 260/280 and 260/230 ratios. Samples were run on a 1% agarose gel with ethidium bromide to rule out RNA degradation. cDNA was then generated from RNA samples with the BioRad iScript Select cDNA Synthesis Kit (#170–8897) in a Thermo Hybaid

Omn-E PCR machine. All cDNA samples were run on a 96-well plate (Applied Biosystems, #4306737) on an Applied Biosystems 7,300 Real-Time PCR system with BioRad iQ Supermix (#170–8862). cDNA-specific PrimeTime qPCR Assay primers (Integrated DNA Technologies) were used for qPCR reactions (http://www.idtdna.com). *Drosophila* GAPDH (*dGapdh*) was used as a housekeeping control. The comparative C(T) method was used to analyze results, as previously described (*Schmittgen and Livak, 2008*). GraphPad Prism six software was used for statistical analyses.

Primers (5'–3'):

## Futsch

Forward: CAAAGCCCACTCACCTTTC
Reverse: CTGCTCCTGCCAACATCT
Probe: AGTCTCTGGAAATGCAGCACCACT

## dNup93-2

Forward: ACACCGTCCGCGAAATAC
Reverse: ACTCAACCGCCACCTTAAC
Probe: ATGGCCGCTGGTTTACTACGGATT

## dNup214

Forward: CCTAAGTGAGGACAAGGATGAG
Reverse: GGCATAGTCTGCAGCTTCTT
Probe: TGCCTTCGACACTTCTACAACGCA

## dNup54

Forward: GAGTGAGCTGACAGAACTCAAG
Reverse: CTCGGCCAGTTTCCGTTTAT
Probe: CCACTGCCACAGCGAAGATACTTGA

## dNup44A

Forward: AAGGTATCCTCCACCAATACCC
Reverse: TTGGGTGCAAACTCCACATC
Probe: TTGTAGACTCGCGGACCAGTGTCA

## dNup62

Forward: CTTGCTGTTGTCTGCATCTC
Reverse: CAGCACCAGCTTCAGGA
Probe: ACATTCTCTTTCGGAACACCGGCA

## Emb (Exportin)

Forward: GGTCACGCGTATGTCATTCA
Reverse: GTTCACATTGACGCCATTCAC
Probe: AGCATGTCCAGATATATGCGGCCC

## dNup43

Forward: TTGCATCTGATCCTCCTCCAC
Reverse: ACCGCCATGGAGTTCGT
Probe: TGGACGTTAAACACGCTGAGATGACC

dGapdh

> Forward: CAACAGTGATTCCCGACCAG
> Reverse: TTCGTCAAGCTAATCTCGTGG
> Probe: CCAAAACTATCGTACAAACCCGGCG

## Rat CCI TBI model

### Subjects and presurgical procedures

Ten adult male Sprague-Dawley rats (Envigo RMS) were pair-housed in standard rat individual ventilated cages under a 12 hr light/dark cycle and maintained at a temperature of 21°C ± 1°C with ad libitum access to food and water. All experimental procedures were approved by the Institutional Animal Care and Use Committee at the University of Pittsburgh. Every attempt was made to limit the number of rats used and to minimize suffering.

### Surgery

TBI was conducted on rats weighing 300–325 g, as previously described (*Sozda et al., 2010*). Briefly, following induction of surgical anesthesia with 4% isoflurane and 2:1 $N_2O:O_2$, rats were intubated, maintained on 2% isoflurane, and mechanically ventilated. A controlled cortical impact (CCI) of moderate severity (2.8 mm tissue deformation at a velocity of 4 m/s) was produced. Immediately after CCI, anesthesia was removed, and the incision was promptly sutured. Rats were returned to home cages, where they remained until being euthanized at 24 hr post-injury.

### Immunohistochemistry

CCI and sham rats were anesthetized intraperitoneally with Fatal-Plus (0.25 mL) and perfused transcardially with 200 mL 0.1 M PBS (pH 7.4), followed by 300 mL 4% paraformaldehyde (PFA) 24 h post-injury. Calvaria were immersion-fixed an additional 24 h in 4% PFA, and brains were subsequently extracted and post-fixed in 4% PFA until dehydration for immunohistochemical staining. Serial dehydrations (increasing concentrations of ethanol, then embedded in paraffin wax) were performed on all brain tissues. After embedding, 5-mm-thick tissues were sectioned at ~1 mm interval.

Slides were selected based on region of interest (ROI), deparaffinized, and rehydrated. ROIs were determined *a priori* as hippocampal regions underlying the injured cortex. Endogenous peroxide was blocked with 3% $H_2O_2$, and slides were subjected to an antigen retrieval step with 1 X Decloaking solution (BioCare Medical, #CB910M). Primary antibody incubation was then performed on all sections for 12 hr at 4°C using mouse anti-Nup62 (IgG), 1:400 (Roche Applied Sciences) or rabbit anti-RanGAP1 (IgG), 1:1000 (Santa Cruz Biotechnology). Slides were washed in 1 X TBS automation buffer (BioCare Medical, #TWB945M) and incubated for 2 hr with secondary goat anti-mouse IgG (1:750, Vector Laboratories, #PK-6102) or anti-rabbit IgG (1:750, Vector Laboratories, #PK-6101). Immunostaining was visualized using a 3,3'-diaminobenzidine (DAB) staining kits (Vector Laboratories, #SK-4100) for ~5 min. Slides were counterstained with aqueous Hematoxylin QS (Vector Laboratories, #H-3404), dehydrated, and cover-slipped using Permount mounting medium (Thermo Fisher Scientific, #SP15-500). Sections were imaged on a Nikon i90 microscope with NS-elements software (Nikon). Nup62 and RanGAP1 pathology was quantified as previously described (*Grima et al., 2017*).

## Human neuropathology

### Subjects

Cases were ascertained from the VA-BU-CLF Brain Bank at Boston University. Detailed inclusion criteria have been described previously (*Mez et al., 2015*). An authorized legal representative provided written consent from brain donation. IRB approval for the brain donation program was obtained through Boston University Alzheimer's Disease & CTE Center and the Edith Nourse Rogers Memorial Veterans Hospital. Neuropathologic evaluation has been described previously (*Cherry et al., 2017*). CTE neuropathologic stage was determined using McKee staging criteria (*McKee et al., 2016*). CTE Stage I or II cases were grouped as mild CTE, and Stage III or IV cases were grouped as severe CTE, as previously published (*Cherry et al., 2016*).

## ELISA

Frozen tissue was collected from identical regions in Broadman area 8/9, as previously described (*Cherry et al., 2016*). Briefly, freshly prepared, ice-cold 5 M guanidine hydrochloride in Tris-buffered saline (20 mM Tris-HCl, 150 mM NaCl, pH 7.4 TBS) containing 1:100 Halt protease inhibitor cocktail (Thermo Fisher Scientific) and 1:100 phosphatase inhibitor cocktail 2 and 3 (Sigma-Aldrich) were added to brain tissues at a 5:1 ratio and homogenized with Qiagen Tissue Lyser LT at 50 Hz for 5 min. Lysates were diluted according to the manufacturer's protocol and centrifuged at 17,000 *g* at 4°C for 15 min. Supernatant was analyzed using a Nup62 ELISA (LSBio, #LS-F22196) according to the manufacturer's protocol. Results are reported as ng/mL.

## Histology

Staining was performed as previously described (*Cherry et al., 2016*). Briefly, tissue blocks of cortical samples were taken from Broadman area 8/9 for all cases. Tissues were embedded in paraffin and cut into 20-µm-thick sections. For chromogenic staining, sections were incubated overnight at 4°C with anti-NUP62 antibody. Sections were treated with biotinylated secondary antibodies and labeled with a 3-amino-9-ethylcarbazol HRP substrate kit (Vector Laboratories). Sections were counterstained with Gill's hematoxylin (Vector Laboratories, #H-3401). For multiplex immunofluorescent images, sections were incubated overnight at 4°C with anti-NUP62 (BD Transduction Laboratories, #610497) and or anti-pTDP43 (Cosmo Bio, #NC0877946) antibodies. Fluorescent labeling was carried out using the PerkinElmer Opal 7-Color Automation IHC kit (#NEL801001KT), per manufacturer's instructions. Sections were counterstained with DAPI.

## Cell culture, transfection, and immunohistochemistry

HEK293T (#CRL-3216) cells were acquired from ATCC, was authenticated and tested negative for mycoplasma contamination. H293T cells were cultured on coverslips coated with poly-L-lysine (Sigma-Aldrich). Cells were transfected with either mRuby, NUP62-mRuby, HA-eGFP, Nup54-HA-eGFP, pEGFP-SF2/SRSF1 (Addgene, #17990), FLAG-NM23-H1/NME1 (Addgene, #25000), Frt-V5-HspB2 (Addgene, #63103), or mRFP-FKBP1A (Addgene, #67514) expression plasmids using TurboFect (Thermo Fisher Scientific), following the manufacturer's instructions. Cells were washed and incubated with primary antibodies overnight: rabbit anti-TDP-43 (10782–2-AP), 1:1000 (Proteintech); or rat anti-phospho TDP-43 (MABN14), 1:1000 (Millipore, Sigma-Aldrich). Cells were washed and incubated with secondary antibodies: anti-rabbit Alexa Flour 647, 1:100 (Life Technologies, #1660844); anti-rat Alexa Flour 568 (Invitrogen, Cat # A-11077). Coverslips were mounted onto slides using Fluoroshield with DAPI (Sigma-Aldrich, #F6057). Cells were imaged with a NIKON A1 eclipse T$_i$ confocal microscope.

# Additional information

## Funding

| Funder | Grant reference number | Author |
|---|---|---|
| National Institute of Neurological Disorders and Stroke | R21 AG064940 | Christopher J Donnelly Udai Bhan Pandey |
| National Institute of Neurological Disorders and Stroke | R01 NS081303 | Udai Bhan Pandey |
| National Institute of Neurological Disorders and Stroke | R21 NS098379 | Christopher J Donnelly Udai Bhan Pandey |
| National Institute of Neurological Disorders and Stroke | U54 NS115266 | Thor Stein |
| National Institute of Neurological Disorders and Stroke | R01 NS084967 | Anthony E Kline |

| Funder | Grant reference number | Author |
|---|---|---|
| National Institute of Neurological Disorders and Stroke | R56 NS113810 | Anthony E Kline |
| National Institute of Neurological Disorders and Stroke | 1R00 NS082376 | Jacob Schwartz |
| Robert Packard Center for ALS Research, Johns Hopkins University | | Udai Bhan Pandey |

The funders had no role in study design, data collection and interpretation, or the decision to submit the work for publication.

## Author contributions

Eric N Anderson, Conceptualization, Data curation, Formal analysis, Investigation, Methodology, Validation, Writing – original draft, Writing – review and editing; Andrés A Morera, Christopher Ebmeier, William Old, Formal analysis, Methodology; Sukhleen Kour, Data curation, Formal analysis, Writing – review and editing; Jonathan D Cherry, Formal analysis, Investigation, Methodology, Resources, Validation, Visualization, Writing – review and editing; Nandini Ramesh, Formal analysis; Amanda Gleixner, Conceptualization, Data curation, Formal analysis, Investigation, Methodology, Validation; Jacob C Schwartz, Conceptualization, Funding acquisition, Project administration, Resources, Supervision, Writing – review and editing; Christopher J Donnelly, Conceptualization, Data curation, Funding acquisition, Project administration, Resources, Supervision, Writing – review and editing; Jeffrey P Cheng, Formal analysis, Investigation, Methodology, Validation; Anthony E Kline, Data curation, Formal analysis, Funding acquisition, Project administration, Resources; Julia Kofler, Data curation, Formal analysis, Investigation, Methodology, Resources, Validation, Visualization, Writing – review and editing; Thor D Stein, Data curation, Formal analysis, Funding acquisition, Methodology, Resources, Supervision, Validation, Writing – review and editing; Udai Bhan Pandey, Conceptualization, Funding acquisition, Project administration, Resources, Supervision, Validation, Writing – original draft, Writing – review and editing

## Author ORCIDs

Christopher Ebmeier (iD) http://orcid.org/0000-0001-7940-6190
Christopher J Donnelly (iD) http://orcid.org/0000-0002-2383-9015
Udai Bhan Pandey (iD) http://orcid.org/0000-0002-6267-0179

## Ethics

All of the animals were handled according to approved institutional animal care and using the approved protocol #21049041 by the University of Pittsburgh.

## Decision letter and Author response

Decision letter https://doi.org/10.7554/eLife.67587.sa1
Author response https://doi.org/10.7554/eLife.67587.sa2

# Additional files

## Supplementary files

• Supplementary file 1. Table summary of the proteomic analysis and GO association of approximately 2000 proteins identified in TBI and control (non-TBI) condition including the fold changes and statistical significance.

• Supplementary file 2. Summary of neuropathology cases (severe CTE and mild CTE) and controls. Table reports all relevant information for the human sample used in the study including age, sex, phosphor TDP-43, dementia, sports played, number of years played, Braak stage (0–6), consortium to establish a registry for Alzheimer's disease (CERAD) score (0–3), and CAA score (0–3).

• Supplementary file 3. GO-ID and p-Values as well as GO associations for genes whose protein levels increase or decrease > 1.6-fold.

• Transparent reporting form

## Data availability

All data generated or analysed during this study are included in the manuscript and supporting files.

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
