## [Decision Letter]

**Acceptance summary:**

Anderson and colleagues use a traumatic brain injury (TBI) model fruit fly to understand the molecular pathogenesis of this disease. They identify proteins that are altered upon TBI, including that Nucleoporins (Nups) are upregulated and this is associated with the aggregation of TDP-43, a protein implicated in neurodegeneration. These findings were confirmed in a TBI rat model and in samples from patients who suffered from TBI. The findings provide molecular insight into TBI pathogenesis and identify a promising therapeutic pathway.

**Decision letter after peer review:**

Thank you for submitting your article "Traumatic injury results in functional impairments in nucleocytoplasmic transport and leads to TDP-43 pathology" for consideration by *eLife*. Your article has been reviewed by 2 peer reviewers, and the evaluation has been overseen by a Reviewing Editor and Huda Zoghbi as the Senior Editor. The following individual involved in review of your submission has agreed to reveal their identity: Shinya Yamamoto (Reviewer #3).

Essential revisions:

As you will see from the reviews below, and from our subsequent discussion, we were enthusiastic about the paper. The reviewers ask a number of questions that are listed below, but there are two items we all agreed on that we hope you will be able to address, possibly by including new experiments or by textual changes:

1. You currently do not directly demonstrate the stability of Nups. We suggest to either reword the text or include a pulse-chase experiment to study the stability of Nups;

2. Clearly discuss the two possible models i.e. TDP-43 aggregation leads to Nup alterations or vice versa; or include experiments that clarify this relationship.

*Reviewer #2:*

Anderson and colleagues investigated the mechanisms underlying toxicity during traumatic brain injury (TBI), which is emerging as a risk factor in several neurodegenerative diseases including Alzheimer's disease, amyotrophic lateral sclerosis (ALS) and chronic traumatic encephalopathy (CTE). They propose deficits in nucleocytoplasmic transport accompanied by TDP-43 pathology to be critical to TBI-induced damage. Starting from a proteomic analysis of brains from a TBI *Drosophila* model previously reported by the team (Anderson et al., 2018 HMG), the authors identify that the majority of the proteins with altered expression (361 proteins) are significantly upregulated, among which nearly half of the 30 known components of the nucleopore complex (NPC) -which the study focuses on. The authors report that repeated trauma provokes aberrant localization and aggregation of specific components of the NPC (including Nup62) in multiple TBI models including *Drosophila*, human cell line, rats, and ultimately in postmortem autopsied brains from CTE patients. Using a functional GFP-reporter assay they demonstrate that TBI impairs active nucleocytoplasmic transport and that pharmacological inhibition of nuclear export improves nuclear/cytoplasmic transport the GFP reporter as well as the fly eclosion deficits caused by TBI.

The team had previously reported that TBI recapitulates TDP-43 pathology in *Drosophila* (Anderson et al., 2018 HMG). Here, the authors report that increased expression of nucleoporins (Nups) including Nup62 in a human cell line (Hek) and in flies induces TDP-43 mislocalization/aggregation, motor impairments and reduced survival, thus leading to the proposal that TBI induced-NPC deficits may provoke TDP-43 pathology.

The authors convincingly demonstrate that TBI provokes morphological and functional alterations in NPC and nucleocytoplasmic transport, which can be partially improved by pharmacological inhibition of nuclear export thus providing mechanistic insight with potential therapeutic avenue. That being said, there are some aspects that need to be clarified to fully support all claims.

1. In figure 1 the authors conclude that trauma increases stability of Nup214, as increased accumulation of Nup214 is observed (by WB) with time of post-traumatic injury. However, the authors also report that mRNA levels are increased and the time course provided does not allow to fully exclude that Nup enhanced levels may occur predominantly at the transcriptional level (rather than at the protein level due to reduced turnover and/or increased stability as claimed by the authors). To substantiate the authors' conclusion on the effects of enhanced stability, pulse-chase/cycloheximide experiments should be considered.

Strengthening such evidence on stability is even more critical given this is in contrast with previously published studies describing a loss (or reduction) rather than increase of NPC components during neurodegenerative processes (Coyne et al., 2020 Neuron; Chou et al. 2017 Nat Neurosc) or aging (D'Angelo et al., 2009 Cell).

2. In their original study describing the TBI *Drosophila* model (Anderson et al. 2018 HMG) the authors had proposed that repetitive trauma induces TDP-43 mislocalization and stress granule formation. Here the authors suggest that TDP-43 pathology may be caused by TBI-induced NPC deficits as cautiously stated "TBI leads to NCT defects, which potentially mediate the TDP-43 pathology in CTE". Given the ample evidence demonstrating that TDP-43 demixing/aggregation induces (in vitro and in vivo) NPC deficits with possible sequestration of some NPC components together with TDP-43 aggregates (Chou et al., 2017 Nat Neuros, Ditsworth et al. 2017 Acta Neuropath, Zhang et al., 2018 Cell, Gasset-Rosa et al. 2019 Neuron), nucleocytoplasmic transport deficits could thus be a consequence of cytoplasmic TDP-43 pathology, rather than the cause. It would be of interest to test whether targeting the NPC rescues/prevents TDP-43 pathology using siRNA-mediated silencing of Nups or pharmacological inhibition, as the authors did to show modest improvement in survival or eclosion, respectively. Providing such evidence could help convincingly determine whether NPC deficits cause TDP-43 pathology in CTE.

3. Line 123: 361 proteins were significantly found to be associated with TBI. Were these consistently found to be impaired with all replicates? The criteria used to select those 361 should be described. This would ascertain their relevance.

4. Figure 2: The sophisticated TBI rat model where trauma seems to be induced unilaterally and provokes TDP-43 pathology in the ipsilateral hemisphere could provide a unique system to test whether TDP-43 aggregates spread with time within the nervous system. It would be of interest to analyze the contralateral side and assess where TDP-43 pathology is also eventually recapitulated.

5. Figure 3 provides elegant evidence for functional rescue of NPT deficits using pharmacological inhibition (using KPT-compound derivatives). Was survival or TDP-43 pathology also improved?

6. There are differences between images shown in Figure 2A: not all cells (stained with Dapi) are Nup positive in the control flies (while it appears to be the case in the TBI larvae). What could be the reason? Are there different cell types represented in control versus TBI?

7. Figure 2B: NPC signal is not consistent between figure 2A and figure 2B (which appears very dotty in the latter), albeit it should be similar: can the authors clarify this discrepancy?

8. Figure 6: the increased intensity of Nup62 IHC signal reported in TBE patients is not fully consistent with that shown by immunofluorescence. Hence the overall IHC background signal is much higher in the patient tissues compared to the single control shown. Including additional controls would strengthen the data.

9, There are some problems with the references. For example: reference 45 (line 121) is not correct here, it should be reference 47. Similarly, on page 134, the references stating that "NPC associated with neurodegenerative" are not correctly cited. They should include Chou et al., 2018; Ditsworth et al. 2017, Zhang et al., Coyne et al., 2020 Neuron, Gasset-Rosa et al., 2019…

10. Figure 2H: the main section of the text refers to arrows/arrowheads that should point at RanGAP1 abnormal staining but they are missing in the figure itself.

11. Line 235: figure citation is not correct. It should be figure 3G (and not Figure 5G).

*Reviewer #3:*

In this study, Anderson and colleagues present data showing that traumatic brain injury (TBI) in *Drosophila* upregulates expression of Nucleoporins (Nups) that in turn impacts nucleocytoplasmic transport (NCT) by taking an untargeted proteomics approach. The authors further explore the effect of Nup upregulation on the aggregation of TDP-43 (TDPH), a protein whose aggregates are a hallmark of several neurodegenerative diseases; importantly chronic traumatic encephalopathy (CTE), a disease that affects those who play high contact sports such as American football. The authors provide convincing data showing that TBI increases Nup expression and affects (likely-pathological) TDP43 aggregation not just in *Drosophila* but consistently in a rat model. They also include data from patient samples to show that increased expression of NUP62 and TDP-43.

Overall, the data presented are very compelling and demonstrative of the power of a multi-species approach (*Drosophila*, rat and human) to explore disease related biology that is translatable and consistent across multiple systems.

1. Line 150 and 414: Here, the authors claim they measured the "Nup214 protein stability". Also in the discussion, they state "In support, our data showed the Nup214 stability and turnover is impaired by repeated traumatic injury in fly brains. It is possible that the slow turnover of Nup214 might be due to dysfunction in protein clearance pathways (e.g. autophagy)." However, what they are doing here is a simple western blot at different time points post-TBI, and it is difficult to tease out differences in protein stability from mRNA/protein synthesis using this strategy. If the authors really want to claim that this is altering protein turnover, one should perform a western upon stopping translation (e.g. cycloheximide application) to claim that the changes that they see are indeed due to alterations in protein stability (not sure how feasible this is in their experimental setting in vivo). Otherwise, they need to do some sort of pulse labeling and tracing experiment. Note that they show that mRNA level of Nup214 (shown as Nup-214 in Fig1I) does increase in adults and larva (Fig1I, K). Hence, both increase in transcription and translation or mRNA stability may all contribute to the increase in Nup214 protein in addition to increased protein stability. If it is difficult to perform these experiments to concretely say that there is a defect in protein stability, the authors should reword their text to say they observe an alteration in "Nup214 protein levels" and not strongly state there is a defect in proteostasis of Nup214 since they don't have direct evidence.

2. Based on the authors model, TBI causes upregulation of certain Nups, which in turn causes TDP-43 aggregation, which then causes neurodegeneration. While the data presented in this paper supports this model, it is also possible that TBI first causes TDP-43 aggregation, which then causes upregulation/aggregation of Nups which then leads to neurodegeneration. I feel the second model is not discussed here and cannot be ruled out since certain experiments are lacking to prefer the first model over the second model. The difference of the two model is whether TDP-43 is upstream or downstream of Nup. For example, if one over-expresses TDP-43 in the absence of TBI, what happens to Nup expression and localization? Can one suppress the short lifespan and motor phenotype caused by over-expressing certain Nup by simultaneously knocking down TDP-43? It is possible both Model 1 and 2 are both correct, and in this case this may suggest that TDP-43 and Nups can regulate each other, causing a vicious cycle in TBI pathogenesis.

---

## [Author Response]

Essential revisions:As you will see from the reviews below, and from our subsequent discussion, we were enthusiastic about the paper. The reviewers ask a number of questions that are listed below, but there are two items we all agreed on that we hope you will be able to address, possibly by including new experiments or by textual changes:1. You currently do not directly demonstrate the stability of Nups. We suggest to either reword the text or include a pulse-chase experiment to study the stability of Nups;

The reviewer made a great point, and we fully agree that a pulse-chase assay would help in clarifying the stability of Nups. We thought about doing a pulse-chase experiment in our fly model of traumatic brain injury (TBI). However, Cycloheximide treatment in *Drosophila* at larval or adult stages leads to lethality (Marco et al., Toxicology Letters, 1982). Moreover, the animal’s post-traumatic injury become symptomatic (Anderson et al., HMG 2018) and we suspect that treating them with cycloheximide would make their phenotypes obviously worst (even lethal) as reported previously, making it difficult to perform this assay in vivo. Therefore, we are rewording our statement about the stability of Nups.

This section now reads “We next asked if trauma alters endogenous nucleoporin protein levels overtime. To address this, we measured Nup214 protein levels in trauma and control larval and adult brains. We performed time-course analysis on larval (0, 2, 4, and 6 hours) or adult (0, 2, 4, 24, and 72 hours) animals’ post-injury to assess Nup214 protein level by Western blot analysis. Interestingly, we found that Nup214 protein levels remain upregulated in both larval and adult brains over the time point examined (Figure 1P-S), suggesting that trauma probably disrupts Nup214 levels and possibly turnover in vivo.”

2. Clearly discuss the two possible models i.e. TDP-43 aggregation leads to Nup alterations or vice versa; or include experiments that clarify this relationship.

This is another great point that the reviewer 3 made. In collaboration with Dr. Christopher Donnelly (senior author), we have a manuscript under review that defined the relationship between TDP-43 aggregation and Nup62 sequestration in iPSC motor neurons, HEK293T cells, and ALS patient postmortem tissues. In order to publicly share our unpublished data while under review, we uploaded our collaborative manuscript titled “Nup62 is recruited to pathological condensates and promotes TDP-43 insolubility in C9orf72 and sporadic ALS/FTLD” on Research Square site (Gleixner et al.,). Here is a link. https://www.researchsquare.com/article/rs^-1^44654/v1

Briefly, in our collaborative manuscript (Gleixner et al., 2021 PREPRINT), we report how TDP43 promotes nuclear depletion and cytoplasmic sequestration of Nup62 in ALS/FTD patient iPSC neurons. We found that formation of TDP-43 containing stress granules promotes the nuclear loss and sequestration of Nup62 in vitro and in vivo. Our data suggests that cytoplasmic Nup62 and TDP-43 interactions are pathological in nature and result in insoluble assemblies in both familial and sporadic ALS/FTD postmortem brain tissues (Gleixner et al., 2021 preprint). More importantly, cytoplasmic Nup62 droplets interaction through the NLS likely alter cytoplasmic TDP-43 phase transition resulting in insoluble inclusions. In addition to this, a recent study has shown that aggregated and pathogenic mutations in TDP-43 promote the sequestration and mislocalization of nucleoporins in mouse primary cortical neurons, human fibroblasts and induced pluripotent stem cell–derived neurons (Chou et al., 2018 Nature Neuroscience). Importantly, nuclear pore pathology has been observed in brain tissue in cases of sporadic ALS and those carrying pathogenic mutations in TDP-43 and C9orf72 (Chou et al., 2018; Cook et al., 2020).

Altogether, the abovementioned studies demonstrate the role of TDP-43 in sequestering nucleoporins in sporadic and familial forms of human neurodegenerative diseases. We are including details about both papers in the Discussion section of our manuscript. This section the discussion now reads:

“Recent data from our and Dr. Donnelly’s lab have demonstrated that cytoplasmic Nup62 droplets exhibit characteristics of proteins that undergo liquid-liquid phase separation, and it is likely that these interaction through the classical NLS promotes deleterious phase transition of cytoplasmic TDP-43 causing it to mature into insoluble inclusions (Gleixner et al., 2021 PREPRINT), providing a mechanism for how Nups might promote TDP-43 pathology in traumatic injury. […] Moreover, the synergistic effects of both mechanisms cannot be ruled out as cause of neurodegeneration in traumatic injury.”

Reviewer #2:[…] The authors convincingly demonstrate that TBI provokes morphological and functional alterations in NPC and nucleocytoplasmic transport, which can be partially improved by pharmacological inhibition of nuclear export thus providing mechanistic insight with potential therapeutic avenue. That being said, there are some aspects that need to be clarified to fully support all claims.1. In figure 1 the authors conclude that trauma increases stability of Nup214, as increased accumulation of Nup214 is observed (by WB) with time of post-traumatic injury. However, the authors also report that mRNA levels are increased and the time course provided does not allow to fully exclude that Nup enhanced levels may occur predominantly at the transcriptional level (rather than at the protein level due to reduced turnover and/or increased stability as claimed by the authors). To substantiate the authors' conclusion on the effects of enhanced stability, pulse-chase/cycloheximide experiments should be considered.Strengthening such evidence on stability is even more critical given this is in contrast with previously published studies describing a loss (or reduction) rather than increase of NPC components during neurodegenerative processes (Coyne et al., 2020 Neuron; Chou et al. 2017 Nat Neurosc) or aging (D'Angelo et al., 2009 Cell).

Please see our response (1 and 2) to the essential comments where we addressed these issues.

2. In their original study describing the TBI *Drosophila* model (Anderson et al. 2018 HMG) the authors had proposed that repetitive trauma induces TDP-43 mislocalization and stress granule formation. Here the authors suggest that TDP-43 pathology may be caused by TBI-induced NPC deficits as cautiously stated "TBI leads to NCT defects, which potentially mediate the TDP-43 pathology in CTE". Given the ample evidence demonstrating that TDP-43 demixing/aggregation induces (in vitro and in vivo) NPC deficits with possible sequestration of some NPC components together with TDP-43 aggregates (Chou et al., 2017 Nat Neuros, Ditsworth et al. 2017 Acta Neuropath, Zhang et al., 2018 Cell, Gasset-Rosa et al. 2019 Neuron), nucleocytoplasmic transport deficits could thus be a consequence of cytoplasmic TDP-43 pathology, rather than the cause. It would be of interest to test whether targeting the NPC rescues/prevents TDP-43 pathology using siRNA-mediated silencing of Nups or pharmacological inhibition, as the authors did to show modest improvement in survival or eclosion, respectively. Providing such evidence could help convincingly determine whether NPC deficits cause TDP-43 pathology in CTE.

As suggested by the reviewer, we fed adult *Drosophila* KPT-350 (0.05mM and 0.5mM or DMSO) for 10 days and found dose-dependent reduction in Tbph aggregation and have incorporate this new data in (Supplementary Figure 4).

3. Line 123: 361 proteins were significantly found to be associated with TBI. Were these consistently found to be impaired with all replicates? The criteria used to select those 361 should be described. This would ascertain their relevance.

The 361 genes are just those whose p-value between TBI and non-TBI is ≤ 0.05. When we did both >1.6-fold up or down and p≤0.05, there were 246 genes that were altered and present “TBI Proteome”. The GO analysis accepted all genes changed >1.6-fold and p ≤ 0.1 to get enough genes to do GO analysis on. These genes were consistently observed in all replicates. We are including these details in the methodology section.

4. Figure 2: The sophisticated TBI rat model where trauma seems to be induced unilaterally and provokes TDP-43 pathology in the ipsilateral hemisphere could provide a unique system to test whether TDP-43 aggregates spread with time within the nervous system. It would be of interest to analyze the contralateral side and assess where TDP-43 pathology is also eventually recapitulated.

We agree with the reviewer. Further studies in the TBI rat model are ongoing and we plan to examine propagation, aggregation and spreading of TDP-43 pathology in different parts of the rat brain over time. It is going to take us several months to perform and analyze these assays in a rodent model. This is beyond the scope of our current manuscript.

5. Figure 3 provides elegant evidence for functional rescue of NPT deficits using pharmacological inhibition (using KPT-compound derivatives). Was survival or TDP-43 pathology also improved?

We agree with the reviewer. Further studies in the TBI rat model are ongoing and we plan to examine propagation, aggregation and spreading of TDP-43 pathology in different parts of the rat brain over time. It is going to take us several months to perform and analyze these assays in a rodent model. This is beyond the scope of our current manuscript.

6. There are differences between images shown in Figure 2A: not all cells (stained with Dapi) are Nup positive in the control flies (while it appears to be the case in the TBI larvae). What could be the reason? Are there different cell types represented in control versus TBI?

All cells stained with DAPI are Nup positive in Figure 2A. Careful analysis of Figure 2A control image will show that the DAPI signals are surrounded by Nup staining albeit the signal is lower in some of the cells. This might be because of antibody penetration issues (technical). We hope the reviewer can appreciate the effort in acquiring these images, as it is difficult to get Nup antibody that stain *Drosophila* tissues uniformly. Furthermore, even within the same *Drosophila* brain tissue there can be some level of variation in the staining pattern of NPC. Hence, we quantify abnormal NPC staining pattern (Figure 2B) in cells located specifically along the midline of the VNC because these cells showed robust and consistent NPC staining.

7. Figure 2B: NPC signal is not consistent between figure 2A and figure 2B (which appears very dotty in the latter), albeit it should be similar: can the authors clarify this discrepancy?

We believe that the reviewer is referring to Figure 2A and 2C. The two images are different in that Figure 2A is only stained with NPC marker Mab414 (imaged in the 568 channel), while Figure 2C is stained with both Mab414 (488 channel, green) and Tbph (568 channel, red) which seems to cause minor alteration in the NPC signal. We hope the reviewer can appreciate that although there was minor effect on the signal when double labeling with Mab414 and Tbph antibodies was done, the abnormal NPC morphology are consistent and still present in Figure 2B

(inset).

8. Figure 6: the increased intensity of Nup62 IHC signal reported in TBE patients is not fully consistent with that shown by immunofluorescence. Hence the overall IHC background signal is much higher in the patient tissues compared to the single control shown. Including additional controls would strengthen the data.

We are including additional high quality IHC images of two controls in our revised manuscript (please see Supplementary Figure 12).

9, There are some problems with the references. For example: reference 45 (line 121) is not correct here, it should be reference 47. Similarly, on page 134, the references stating that "NPC associated with neurodegenerative" are not correctly cited. They should include Chou et al., 2018; Ditsworth et al. 2017, Zhang et al., Coyne et al., 2020 Neuron, Gasset-Rosa et al., 2019…

We apologize for this error. We have checked every reference carefully and corrected them.

10. Figure 2H: the main section of the text refers to arrows/arrowheads that should point at RanGAP1 abnormal staining but they are missing in the figure itself.

We have added the arrows/arrowheads in the figure. Thanks for pointing out this mistake.

11. Line 235: figure citation is not correct. It should be figure 3G (and not Figure 5G).

We have corrected the figure citation.

Reviewer #3:[…] 1. Line 150 and 414: Here, the authors claim they measured the "Nup214 protein stability". Also in the discussion, they state "In support, our data showed the Nup214 stability and turnover is impaired by repeated traumatic injury in fly brains. It is possible that the slow turnover of Nup214 might be due to dysfunction in protein clearance pathways (e.g. autophagy)." However, what they are doing here is a simple western blot at different time points post-TBI, and it is difficult to tease out differences in protein stability from mRNA/protein synthesis using this strategy. If the authors really want to claim that this is altering protein turnover, one should perform a western upon stopping translation (e.g. cycloheximide application) to claim that the changes that they see are indeed due to alterations in protein stability (not sure how feasible this is in their experimental setting in vivo). Otherwise, they need to do some sort of pulse labeling and tracing experiment. Note that they show that mRNA level of Nup214 (shown as Nup-214 in Fig1I) does increase in adults and larva (Fig1I, K). Hence, both increase in transcription and translation or mRNA stability may all contribute to the increase in Nup214 protein in addition to increased protein stability. If it is difficult to perform these experiments to concretely say that there is a defect in protein stability, the authors should reword their text to say they observe an alteration in "Nup214 protein levels" and not strongly state there is a defect in proteostasis of Nup214 since they don't have direct evidence.

We have addressed this issue in more details in the “essential comment” section

(comment 1). We agree with the reviewer and we are re-wording our statement about the Nup62 protein stability and toning it down. Specifically, we have removed the word “stability” and replaced it with “Nup214 protein levels” as suggested by the reviewer.

Due to technological obstacles, we are unable to perform a pulse chase experiment in vivo. We agree that the Nup214 mRNA levels are also increased suggesting that both TBI impacts nucleoporins at transcription and translational levels. We have added a statement addressing both possibilities in the discussion (see response to comment #15).

2. Based on the authors model, TBI causes upregulation of certain Nups, which in turn causes TDP-43 aggregation, which then causes neurodegeneration. While the data presented in this paper supports this model, it is also possible that TBI first causes TDP-43 aggregation, which then causes upregulation/aggregation of Nups which then leads to neurodegeneration. I feel the second model is not discussed here and cannot be ruled out since certain experiments are lacking to prefer the first model over the second model. The difference of the two model is whether TDP-43 is upstream or downstream of Nup. For example, if one over-expresses TDP-43 in the absence of TBI, what happens to Nup expression and localization? Can one suppress the short lifespan and motor phenotype caused by over-expressing certain Nup by simultaneously knocking down TDP-43? It is possible both Model 1 and 2 are both correct, and in this case this may suggest that TDP-43 and Nups can regulate each other, causing a vicious cycle in TBI pathogenesis.

The reviewer made a valid point. We remain unbiased and open about both models.

It has been previously reported that overexpression of a C-terminal fragment of TDP-43 (TDP43-CTF) itself in primary cortical neurons perturbed Nup98 localization and distribution. This is further evident from mAb414 staining showing that TDP43-CTF overexpression is sufficient to cause FG Nup pathology (Chou et al., 2018). These findings suggesting a cascade of events due to defects in nucleocytoplasmic transport processes. We plan to follow up on the experiments that the reviewer suggested to directly assess which mechanism(s) are involved and which model is more relevant to TBI-mediated neurodegeneration. We are including statements about both models in the Discussion section (see response to essential comment #2).